# IKEA Manuals at Work: 4D Grounding of Assembly Instructions on Internet Videos

**Yunong Liu**[1]    **Cristobal Eyzaguirre**[1]    **Manling Li**[1]    **Shubh Khanna**[1]
**Juan Carlos Niebles**[1]    **Vineeth Ravi**[2]    **Saumitra Mishra**[2]    **Weiyu Liu**[1][*]    **Jiajun Wu**[1][*]

[1]Stanford University    [2]J.P. Morgan AI Research

## Abstract

Shape assembly is a ubiquitous task in daily life, integral for constructing complex 3D structures like IKEA furniture. While significant progress has been made in developing autonomous agents for shape assembly, existing datasets have not yet tackled the 4D grounding of assembly instructions in videos, essential for a holistic understanding of assembly in 3D space over time. We introduce IKEA Video Manuals, a dataset that features 3D models of furniture parts, instructional manuals, assembly videos from the Internet, and most importantly, annotations of dense spatio-temporal alignments between these data modalities. To demonstrate the utility of IKEA Video Manuals, we present five applications essential for shape assembly: assembly plan generation, part-conditioned segmentation, part-conditioned pose estimation, video object segmentation, and furniture assembly based on instructional video manuals. For each application, we provide evaluation metrics and baseline methods. Through experiments on our annotated data, we highlight many challenges in grounding assembly instructions in videos to improve shape assembly, including handling occlusions, varying viewpoints, and extended assembly sequences.

## 1 Introduction

The autonomous assembly of complex 3D structures requires an understanding at multiple levels of abstraction. The top level is task planning—decomposing the task into subtasks and computing their dependency and ordering, as outlined in an instruction manual; the middle level is visual grounding—registering each part with perceptual input at the pixel level, identifying their geometry and pose, so that the manual instructions get translated to actionable steps; the bottom level is motion planning and control—executing the steps based on the instructions and the identified states, avoiding collisions given the specific embodiment. A successful assembly requires solving all of these problems together, which is difficult. Thus, assembly remains a significant challenge in AI and robotics.

When developing assembly benchmarks, researchers often choose IKEA furniture as target objects due to its ubiquity and standardization. However, existing benchmarks and datasets focus only on part of the assembly problem. Classic work on designing assembly instructions provides step-by-step guidance for task planning, but they are not visually grounded [1]; recent work has attempted to ground assembly steps to 3D part models and register them at the pixel level [2], but it does not include action trajectories of how the assembly may actually happen; datasets that provide 3D groundings in the form RGB-D videos are collected in controlled lab environments [3, 4]; a video dataset of IKEA furniture assembly from the Internet includes more diverse demonstrations [5], but it lacks correspondence with 3D part models and alignment with the instruction manuals.

To address these limitations, we introduce the IKEA Video Manuals dataset, a large-scale multimodal dataset with high-quality, spatial-temporal alignments of step-by-step instructions, 3D models, and

---

[*]Equal advising. Project page: yunongliu1.github.io/ikea-video-manual

38th Conference on Neural Information Processing Systems (NeurIPS 2024) Track on Datasets and Benchmarks.

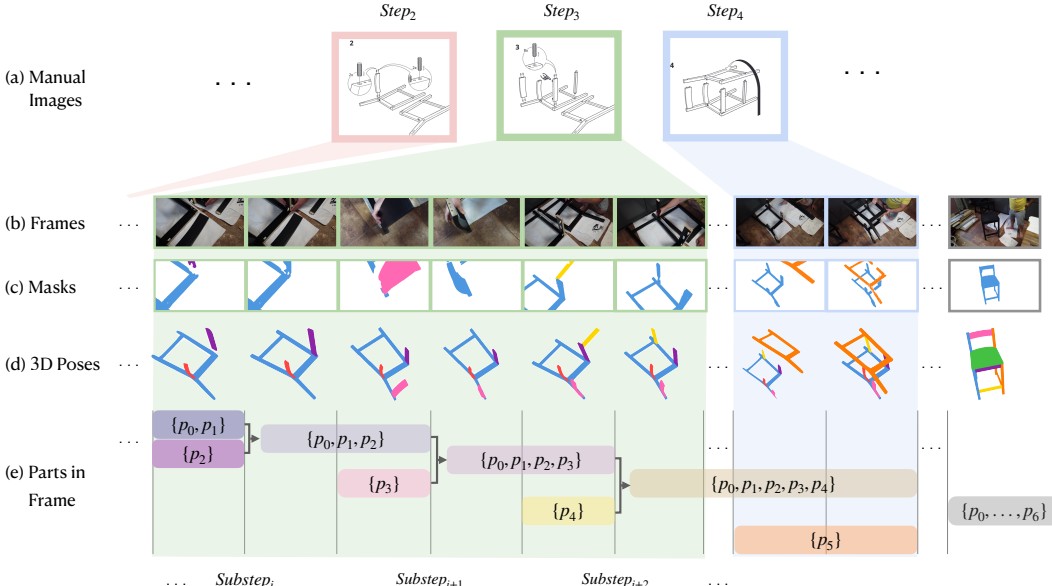

Figure 1: **Dataset Overview.** (a) Manual images showing the assembly steps. (b) Video frames from the corresponding assembly videos. Temporal alignment between the video frames and each assembly step is also provided. (c) Segmentation masks for individual parts and sub-assemblies that are being constructed in each frame. When two parts are assembled, their masks are combined. (d) 6-DoF poses for parts and sub-assemblies in each frame. (e) Tracking of individual parts and sub-assemblies across video frames, capturing the frame-by-frame assembly process.

real-world video demonstrations from the Internet. IKEA Video Manuals provides 34,441 annotated video framesfrom 98 assembly videos for 6 furniture categories. We provide extensive video annotations, including 2D-3D part correspondences, temporal step alignments, and part segmentation. The key contributions of our work include:

- A novel multimodal dataset built on top of Internet videos to capture the complexity and diversity of real-world furniture assembly;

- Comprehensive annotations, including 2D-3D part correspondences, temporal step alignments, and part segmentation;

- Extensive experiments on plan generation, part segmentation and pose estimation, video object segmentation, and part assembly with the annotated data.

## 2 Related Work

**Instructional Video Datasets and Procedural Understanding.** Instructional video datasets are essential for advancing the understanding and learning of procedural tasks. Existing datasets such as YouCook2 [6], COIN [7], and EPIC Kitchens [8] focus on cooking and daily activities, while datasets such as IKEA ASM [3], IKEA-FA [9, 10], and Assembly101 [11] target shape assembly. However, these datasets are predominantly annotated at the coarse level with action labels, limiting their utility for grounding procedural tasks in 3D. Instructional datasets have led to the development of various methods for procedural understanding, including action recognition, video classification, action segmentation, localization, and prediction [7, 12, 13]. To capture the hierarchical nature of procedural tasks, some methods utilize hand-centric features and script data [14]. Temporal and semantic relationships between actions in complex activities have been explored using graph-based representations [15–17]. Techniques such as unsupervised learning from narrated instruction videos aim to identify key procedural steps [18], and cross-task weak supervision has been introduced to improve transfer learning of step localization [12]. Despite advances in procedural understanding, current efforts are mostly limited to 2D video data without grounding in 3D. This absence of 3D context restricts the learning of spatial relations and object interactions essential to real-world task understanding. In this paper, we address these limitations by focusing on the 4D grounding of

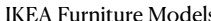

IKEA Furniture Models          Environments

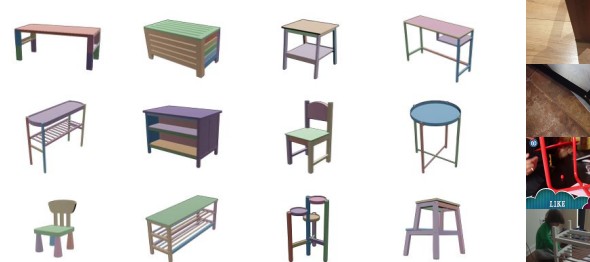
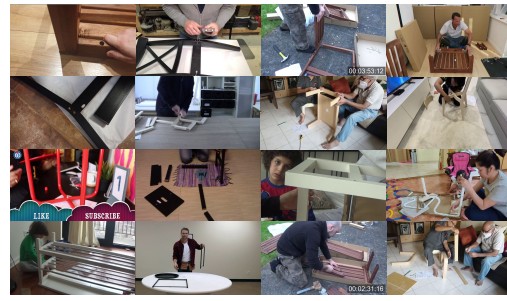

Figure 2: **Dataset Diversity.** (a) Examples of 3D furniture models across different categories in our dataset, showing structural and functional variety. (b) Diverse assembly environments from our video collection, demonstrating real-world complexity including different lighting conditions, camera angles, and backgrounds. The diversity of both furniture types and environments presents unique challenges for grounding.

assembly plans in videos. We provide a more comprehensive comparison with existing shape assembly datasets in Section 3.3.

**Shape Assembly.** Shape assembly has been a long-standing research problem, with foundational works focusing on constructing 3D shapes by assembling parts from repositories [19–22]. More recent approaches can generate parts and predict their transformations to obtain a new shape [23–25]. Building on PartNet [26], graph-based learning methods like DGL [27] and RGL [28] predict the 6-DoF poses of individual parts to construct a shape, while approaches such as IET [29] leverage the transformer neural network to model part relationships. However, these methods ignore potential guidance for assembly from different forms of instructions, limiting their applicability to more complex shapes. To address this gap, datasets like IKEA-Manual [2] have been introduced, providing 3D IKEA furniture models and corresponding instruction manuals. Despite the advancement, existing datasets do not fully capture the complexity and diversity of real-world assembly processes. IKEA Video Manuals addresses these limitations by introducing a multimodal dataset that aligns real-world assembly videos with 3D models and human-designed visual manuals.

## 3 Grounding Assembly Instructions on Internet Videos

In this section, we formally define the spatio-temporal data involved in grounding assembly instructions. We then present the features and analysis of our IKEA Video Manuals dataset.

### 3.1 Definition

As illustrated in Fig. 1, we provide a dataset of grounded assembly instructions for IKEA furniture. Each piece of furniture $S$ in the dataset consists of a set of **3D parts** $\{p_1, ..., p_N\}$, where $N$ denotes the number of parts. The 6-DoF poses of the 3D parts are denoted by $\{\zeta_1, ..., \zeta_N\}$. The furniture is assembled by transforming the 3D parts according to the poses, i.e., $S = \{\zeta_1(p_1), ..., \zeta_N(p_N)\}$. In our dataset, each video consists of a sequence of **frames** $\{f_1, ..., f_T\}$ and demonstrates a physically realistic assembly process (Fig. 1b). In the assembly process, 3D parts are combined, and new sub-assemblies are formed. We denote each **sub-assembly** by $A$, which can be an individual part or a set of previously combined parts. In each frame, we identify the sub-assemblies that are being constructed (Fig. 1e). We further localize each sub-assembly in each frame of the video with a **segmentation mask**, which assigns pixels to their associated sub-assembly (Fig. 1c). A **6-DoF pose** of each sub-assembly in the camera's coordinate frame is included along with the camera intrinsic parameters (Fig. 1d). Combining the poses of the furniture parts throughout the video gives rise to the **4D grounding** of the assembly video.

To provide high-level guidance of the assembly procedure, we also include the **instruction manual** for each piece of furniture in our dataset. Each instruction manual consists of a sequence of $L$ images $\{m_1, ..., m_L\}$ (Fig. 1a). Similarly to the videos, each image from the manual is also annotated with the identities, masks, and poses of the appeared sub-assemblies. We observe that instruction manuals often provide higher-level illustrations of assembly procedures than the assembly videos and omit details of the part trajectories. To associate the two types of instructions for 3D assembly, a temporal alignment between them is established as a mapping $\phi(m_i) \rightarrow \{f_j, .., f_k\}$, where $j \leq k$.

### 3.2 Key Features

**Multiple Instruction Forms for Assembly.** Shape assembly is a complex task that requires geometric reasoning and planning; our dataset provides different types of instructions to facilitate the process, including high-level assembly tree, instruction manuals, and assembly videos. The high-level assembly tree omits geometric and spatial information about parts but provides the decompositions of the assembly process into smaller steps. Manuals provide pictorial illustrations of the assembly process. These illustrations provide both a coarse breakdown of the assembly process and relative poses between object parts in 2D. Compared to instruction manuals, assembly videos provide a more fine-grained assembly process where parts are assembled one at a time, and the whole trajectory of the object movement till the construction of each sub-assembly is shown.

**Temporal Alignment of Instruction and Assembly Process.** Different instruction forms provide different temporal decomposition of the process. Instruction manuals often provide high-level decomposition, while how-to videos demonstrate more detailed steps of each part assembly. Our dataset aligns each step from the instruction manual with a sequence of substeps, in which sub-assemblies are formed (Fig. 1a and Fig. 1b). These substeps are further mapped to segments of the how-to videos, which provide a frame-by-frame demonstration of the assembly.

**Spatial Alignment of Instruction and Assembly Process.** Our dataset further provides spatial details of the whole assembly process in 3D observed from the instruction manuals and videos. These details are provided in the form of 6-DoF pose trajectories of the furniture parts. Specifically, for each video frame, the parts being assembled are annotated with a 2D image mask. The pose of each object part in the camera frame is provided, while the relative poses between parts that are being assembled are detailed. With the additional camera intrinsics we provide, the 3D parts can also be projected into 2D image space and aligned with their corresponding 2D mask.

**Diversity in Assembly.** As shown in Fig. 2, our dataset captures a wide range of furniture assembly scenarios from the Internet videos, encompassing various furniture types, designs, and assembly processes. The dataset includes six main furniture classes. Different instances of the same furniture class are also provided to present the difference in assembly due to designs, sizes, and structures. Furthermore, the dataset also includes multiple assembly videos for each instance. These videos capture various perspectives, environments, and individuals performing the assembly, allowing an in-depth analysis of the variability and commonalities in assembly processes.

**Complexity in Real World Videos.** By including assembly videos from the Internet, our dataset captures the complexity inherent in real-world data. The assembly videos present visual challenges such as changes in camera parameters, camera movements, diverse environment backgrounds, and heavy occlusions. These challenges are representative of the difficulties encountered in practical applications and provide a meaningful benchmark for evaluating the performance and robustness of assembly understanding algorithms.

### 3.3 Comparison with Existing Datasets

Our dataset is the first to provide 6-DoF pose annotations for furniture assembly from internet videos, capturing real-world complexity across diverse settings. We compare it to existing assembly datasets in Table 1. Unlike prior video datasets with 3D information, our dataset features a significantly wider variety of objects (36 furniture types comprising 268 parts) and environments (over 90 different settings). Our dataset uniquely captures variations in the assembly process for each piece of furniture, with 25% of items having multiple valid assembly sequences, the Laiva shelf having the highest number with eight variations. Our data collection pipeline addresses key challenges in real-world video annotation, notably ensuring consistent camera parameters and maintaining accurate relative poses between parts across frames. Additional comparisons, including action labels and human pose information, are discussed in Appendix I.

### 3.4 Dataset Statistics and Analysis

**Statistics.** Our dataset comprises 98 RGB videos. For each video, we annotated 1 frame per second, resulting in a total of 34441 annotated frames. On average, 316 frames are annotated for each video. In total, the dataset contains 137 high-level assembly steps from instructional manuals and 1120 detailed substeps from videos. The dataset provides assembly instructions for 36 unique IKEA furniture models, including 20 chairs, 8 tables, 3 benches, 1 desk, 1 shelf, and 3 other categories.

**Analysis of Assembly Process.** Our dataset includes the assembly process for complex 3D structures. On average, each piece of furniture has seven parts. The dataset also captures the variability in

Table 1: **Comparison with Existing Assembly Datasets.** Our dataset uniquely provides 6D pose annotations on internet videos, capturing furniture assembly in diverse, real-world settings. *While camera parameters in both our dataset and IKEA-Manual are estimated, we implement additional processing to ensure consistent parameters within each video segment, provided there are no obvious camera changes.

| Dataset | # Object Class | # Object | Video Source | # Environment | 3D Object Model | 3D Info. | Camera Param. |
|---|---|---|---|---|---|---|---|
| IKEA Video Manuals (Ours) | 6 | 36 | Internet | ∼90 | ✓ | 6-DoF Pose | Estimated* |
| Assembly101 [11] | 15 | 101 | Lab | 1 | ✗ | Depth | Calibrated |
| HA-ViD [30] | 1 | 35 parts | Lab | 1 | ✓ | Depth | Calibrated |
| IKEA-Manual [2] | 6 | 102 | / | / | ✓ | 6-DoF Pose | Estimated |
| IKEA Ego 3D [31] | 4 | 4 | Lab | 1 | ✗ | Depth | Calibrated |
| IKEA ASM [3] | 3 | 4 | Lab | 5 | ✗ | Depth | Calibrated |
| tIKEA in the Wild [32] | 14 | 420 | Internet | ∼1000 | ✗ | / | Uncalibrated |

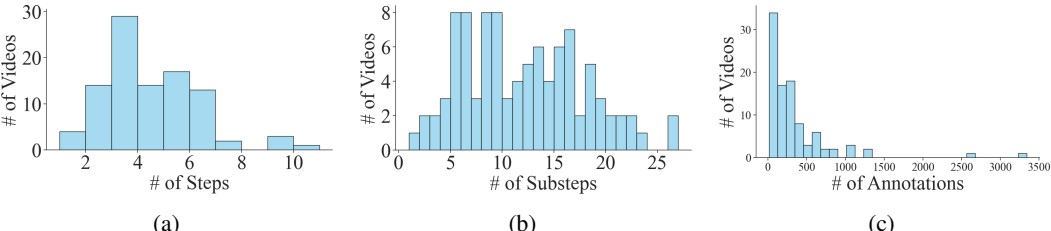

Figure 3: **Dataset Statistics.** (a) Distribution of the number of assembly steps in videos. (b) Distribution of the number of sub-assembly steps (substeps) in videos. (c) Distribution of the number annotations in videos.

assembly complexity, with the longest video spanning 49 minutes and the average duration per video being six minutes. Fig. 3a shows the distribution of the number of steps in the videos. Fig. 3b presents the distribution of the number of substeps in the videos. Fig. 3c illustrates the distribution of the number of mask and pose annotations in the videos. On average, each step from the instruction manual corresponds to eight substeps in the videos.

# 4 Data Collection and Annotation

The IKEA Video Manuals dataset is a comprehensive multimodal dataset that aligns 3D furniture, step-by-step instructions, and real-world video demonstrations. We collect data from various sources and perform extensive annotations to provide a rich resource for grounding furniture assembly instructions. In this section, we describe the data sources, temporal annotations, mask annotations, and pose annotations. We provide details and illustrations of our interface in Appendix C and E.

## 4.1 Collecting 3D Models and Assembly Videos

Building on the IKEA-Manual dataset [2] and IAW dataset [32], we collect 36 segmented 3D furniture models from the IKEA-Manual dataset and 98 assembly videos associated with them in the IAW dataset, providing temporal alignment between instruction steps and video segments. as illustrated in Fig. 4a. We focus on collecting fine-grained temporal segmentation and pose annotations for videos.

## 4.2 Annotating Temporal Segmentation and Part Identity

In our dataset, we provide fine-grained temporal segmentation of the videos based on the construction of each sub-assembly in the assembly processes. To annotate such data, we first extract video segments for each manual step from the IAW dataset. As a single step in the instruction manual often involves combining multiple furniture parts, we further decompose each video segment into smaller intervals, which we call substeps (shown in Fig. 4b). In these substeps, a new sub-assembly can be constructed or deconstructed. For each substep, we sample video frames at 1 FPS. As illustrated in Fig. 4c, we manually annotate the identities of the furniture parts in the first frame for each substep. This annotation is essential for ensuring consistent mask and pose annotations in the following stages because many furniture parts can be similar in appearance (e.g., the legs for a table).

## 4.3 Annotating Segmentation Masks

To track the identities of the furniture parts throughout an assembly video, we annotate 2D image segmentation masks for 3D parts in the sampled frames. To facilitate the annotation process, we develop a web interface that displays auxiliary 2D and 3D information and enables interactive mask annotation based on the Segment Anything Model (SAM) model [33]. For each target part, an

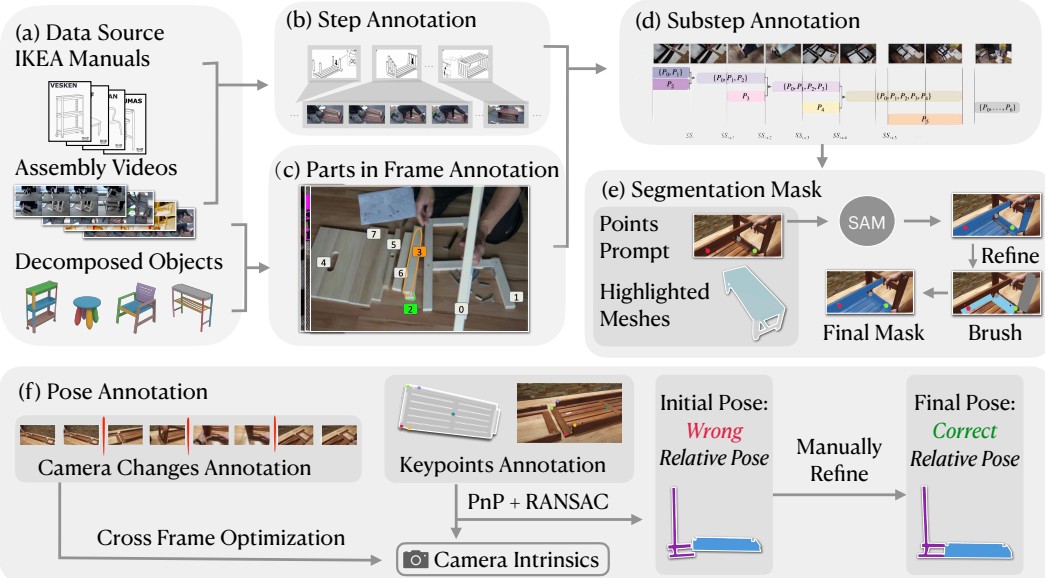

Figure 4: **Data Collection and Annotation Pipeline.** (a) Collecting 3D furniture models, associated assembly manuals and videos. (b) Annotating coarse temporal segmentation of videos into segments showing each assembly step. (c) Tracking identities of 3D parts throughout each video keyframe. (d) Fine-grained temporal segmentation into substeps showing the construction of each sub-assembly. (e) Annotating 2D segmentation masks for parts and sub-assemblies in sampled frames using an interactive interface powered by the SAM model. (f) Estimating camera parameters and 6D part poses in each frame using 2D-3D correspondences, PnP, RANSAC, and manual refinement.

annotator sees the 3D furniture model with the target part highlighted and the part's 2D location in the first frame of the current substep (as seen in Fig. 4e). The annotator then labels the keypoints in the current frame, which is fed into the SAM model to generate the 2D segmentation mask in real-time. Modifications can be easily made by adding or removing key points. To deal with SAM's inherent limitations in extracting boundaries between parts with similar textures or in low-light regions, we further allow annotators to manually adjust the mask annotations with a brush and an eraser tool.

### 4.4 Annotating 2D-3D Correspondence

Inferring relative poses between 3D furniture parts from 2D videos is important for extracting grounded assembly knowledge from videos. We annotate 3D poses of furniture parts in the sampled video frames. Manual annotation is essential as real-world assembly videos often feature occlusions, challenging viewpoints, and partial visibility - these are difficult cases for recovering poses directly from depth estimation. A key challenge is to ensure the 3D trajectories of object parts respect the geometric constraints enforced by the physical assembly process (e.g., the final relative pose between two parts inferred from a video needs to align with their poses in the 3D furniture model). Besides minimizing the projection error of 3D parts in 2D images, we additionally emphasize the accuracy of relative poses between furniture parts and cross-frame consistency in the annotation process.

A prerequisite for achieving spatially and temporally accurate pose annotation is a correct estimation of camera parameters from the video. As illustrated in Fig. 4f, we first identify video frames where potential changes in camera intrinsics occur (e.g., due to focal length adjustments or switching between multiple cameras). We then annotate 2D-3D point correspondences between the 3D models and their 2D projections in the video frames. Using a combination of these two types of annotations, we estimate camera intrinsics for each video. In particular, for each estimated candidate intrinsics, we use the Perspective-n-Point (PnP) algorithm [34] to estimate the pose of the object. We then select the intrinsic that minimizes the reprojection errors for frames between two camera changes. To further refine the camera intrinsics, we apply the Random Sample Consensus (RANSAC) [35] algorithm to filter out outliers in the annotated keypoints. From the resulting top ten intrinsics in each video segment, we choose the one that provides a minimal set of camera intrinsics.

We develop an interactive interface for refining pose annotations initially estimated from the 2D-3D point correspondences. The interface allows annotators to control the virtual camera using axis-

Table 2: **Assembly Plan Generation Results.** Results on the IKEA Video Manuals dataset compared to the IKEA-Manual dataset. Two heuristic baselines are evaluated using Simple Matching and Hard Matching criteria. Precision, Recall, and F1 scores are reported for each setting.

| Method | Dataset | Simple Matching | | | Hard Matching | | |
|---|---|---|---|---|---|---|---|
| | | Precision | Recall | F1 Score | Precision | Recall | F1 Score |
| SingleStep | IKEA-Manual | 100.00 | 35.77 | 48.64 | 10.78 | 10.78 | 10.78 |
| GeoCluster | IKEA-Manual | 44.90 | 48.46 | 43.53 | 16.54 | 16.50 | 16.30 |
| SingleStep | Ours | 98.98 | 16.86 | 26.88 | 3.06 | 2.55 | 2.72 |
| GeoCluster | Ours | 43.04 | 24.16 | 29.74 | 14.98 | 9.49 | 11.51 |

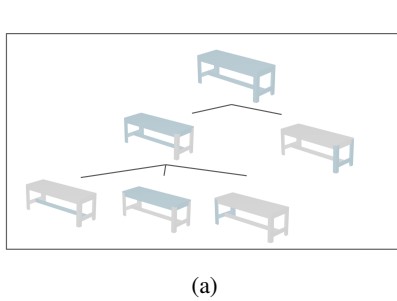
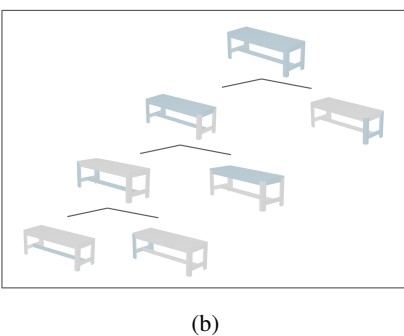

(a)                                    (b)

Figure 5: **Example of Hierarchical Assembly Trees.** (a) Assembly tree structure derived from the high-level steps in the IKEA manual. (b) More detailed assembly tree structure extracted from the fine-grained substeps annotated in the assembly videos.

aligned controls, enabling them to view the 3D scene from different orthographic perspectives. This helps identify and correct errors in relative part poses that may be difficult to detect in a single rendered image. The annotators are able to use the interface to refine the part poses by rotating and translating parts in 3D space. Annotators refine part poses by manipulating them in 3D space, comparing the real-time 3D view with corresponding video frames. To further improve the accuracy of relative poses, parts that appear together in a video frame are annotated together with a visualization of their 3D locations. Temporally smoothness of the part trajectories is improved by initializing part poses with poses from the previous frame.

## 5   Applications

We evaluate existing methods on five tasks essential for grounding instructional assembly videos. We use pretrained models without additional finetuning on our dataset.

### 5.1   Assembly Plan Generation

Assembly plan generation aims to predict a hierarchical assembly plan from a sequence of video frames $\{f_1, \ldots, f_T\}$ depicting a furniture assembly process. The predicted plan is represented as a directed acyclic graph $\mathcal{G} = (\mathcal{V}, \mathcal{E})$, where each node $v \in V$ corresponds to a subset of $K$ parts $\{p_1, p_2, \ldots, p_K\}$, and each edge $e \in \mathcal{E}$ indicate assembly order and parent-child relationships. The root node $v_r$ represents the final assembled shape. IKEA manuals present assembly instructions as diagrams, often combining multiple parts in one step (Fig. 5a). Our IKEA Video Manuals presents physically realistic assembly plans extracted from Internet videos (Fig. 5b).

**Experiment Setup.** We consider two heuristic baselines from IKEA-Manual [2]. The first baseline, SingleStep, constructs an assembly tree with all parts directly connected to the root node, corresponding to assembling all parts in a single step. The second baseline, GeoCluster, uses a pre-trained DGCNN [36] to extract 3D features for each part and constructs the assembly tree iteratively by grouping geometrically similar parts in individual steps. We report the precision, recall, and F1 score based on two matching criteria between the predicted and ground truth plans, as proposed in [2]. Simple Matching considers a predicted node as correct if it matches a ground truth node based on the primitive parts. Hard Matching requires the predicted node to match the ground truth node based on both the parts and parent-child relationships.

Table 3: **Part-conditioned Segmentation Results** on the IKEA Video Manuals dataset. IoU and Top-5 IoU metrics are reported.

| Method | IoU | Top-5 IoU |
|--------|-----|-----------|
| CNOS [37] | 0.09 | 0.21 |
| SAM-6D [38] | **0.16** | **0.40** |

Table 4: **Part-conditioned 6D Pose Estimation Results** on the IKEA Video Manuals dataset using the ADD and ADD-S metrics.

| Method | ADD | ADD-S |
|--------|-----|-------|
| SAM-6D [38] | 2.34 | 1.85 |
| MegaPose [39] | **1.36** | **0.89** |
| Diff. Rendering (MSE) | 3.33 | 2.91 |
| Diff. Rendering (Occlusion-Aware) | 3.29 | 2.86 |

**Results and Analysis.** As shown in Table 2, both baselines struggle to generate accurate plans that resemble the assembly process seen in the videos. Utilizing the geometric features of the furniture parts, GeoCluster slightly outperforms SingleStep. Since video-derived assembly plans are often longer and presents greater diversity than plans extracted from instruction manuals, both models perform worse on our dataset than the IKEA-Manual dataset.

## 5.2 Part-Conditioned Segmentation

This task leverages the diverse videos in the IKEA Video Manuals dataset to evaluate the performance of part segmentation methods in real-world scenarios. The part-conditioned segmentation task requires the models to predict pixel-wise segmentation masks for the furniture parts seen in the assembly videos. Formally, given a frame $f$ and a sub-assembly $A$, the goal is to predict a binary segmentation mask for the sub-assembly.

**Experimental Setup.** We test pre-trained CNOS [37] and SAM-6D [38]. Both models can segment novel objects given their 3D models. In total, we evaluated on 12296 examples from the dataset. We only include sub-assemblies that are unique in shape to remove ambiguities. We report Intersection-over-Union (IoU) and Top-5 IoU.

**Results and Analysis.** As shown in Table 3, both CNOS and SAM-6D obtain relatively low performance on the IKEA Video Manuals dataset. We hypothesize that SAM-6D outperforms CNOS by considering additional geometric features, including shape and size. Common failures of both models stem from heavily occluded parts, visually complex backgrounds, and textureless 3D shapes. The results highlight the existing challenges in detecting object parts in Internet videos.

## 5.3 Part-Conditioned Pose Estimation

Estimating 3D poses of furniture parts from each video frame is essential for grounding the assembly process. Given a video frame $f$ and a furniture sub-assembly $A$, the goal of part-conditioned pose estimation is to predict the 6-DoF pose of the sub-assembly $A$ in the frame $f$.

**Experimental Setup.** We sample 7795 annotations from the IKEA Video Manuals dataset and use ground truth masks for evaluation. We evaluate four methods: SAM-6D [38], MegaPose[39], and two differentiable rendering-based methods. SAM-6D requires a depth image in addition to the RGB image, which we obtain using the depth estimation model MiDaS [40]. The first differentiable rendering method uses Mean Squared Error (MSE) loss as the optimization objective, while the second incorporates an occlusion-aware silhouette re-projection loss proposed in PHOSA [41]. Both differentiable rendering methods use 20 random initial poses, refine the top five candidates with the lowest initial loss for 500 iterations, and output the pose with the lowest final loss. We report the ADD and ADD-S metrics, which are commonly used in the 6D pose estimation literature [42].

**Results and Analysis.** Table 4 presents the quantitative results of all four methods on the IKEA Video Manuals dataset. While MegaPose achieves better performance overall, all four methods struggle with real-world challenges such as partial visibility and occlusions. MegaPose particularly struggles with symmetric parts and challenging viewpoints, while SAM-6D's performance is limited by the accuracy of the depth estimation in complex scenes. In Appendix H, we provide detailed error analysis and examples of failure cases. The high ADD and ADD-S scores indicate that the predicted poses are far from the ground truth poses, highlighting the challenges posed by the IKEA Video Manuals dataset.

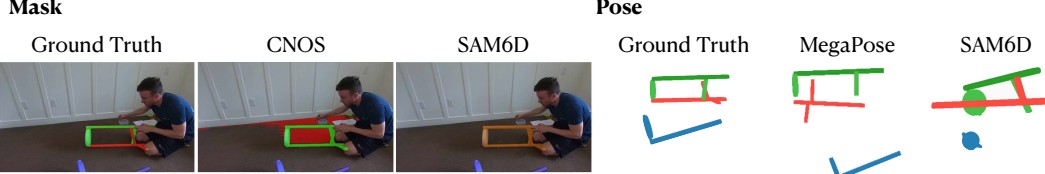

| Mask | | | Pose | | |
| Ground Truth | CNOS | SAM6D | Ground Truth | MegaPose | SAM6D |

Figure 6: **Qualitative Examples.** Examples of part-conditioned segmentation (left) and part-conditioned pose estimation (right) on the IKEA Video Manuals dataset. For segmentation, the ground truth masks are shown along with predicted masks from CNOS [37] and SAM-6D [38]. The orange mask shown in the SAM-6D result is caused by the overlap of red and blue masks. For pose estimation, the ground truth 6D pose is shown along with predicted poses from SAM-6D and MegaPose. [39]

Table 5: **Video Object Segmentation Results** on the IKEA Video Manuals dataset compared to other benchmark datasets. J&F scores are reported for each method.

| Method | Ours | SA-V [43] | MOSE [44] | DAVIS 2017 [45] | LVOS [46] | YTVOS 2019[47] |
|---|---|---|---|---|---|---|
| SAM2 Hiera-L [43] | 73.6 | 75.6 | 77.2 | 91.6 | 76.1 | 89.1 |
| Cutie-base [48] | 54.7 | 60.7 | 69.9 | 87.9 | 66.0 | 87.0 |

## 5.4 Video Object Segmentation

Video object segmentation in our dataset focuses on tracking individual furniture parts within assembly substeps. Given a video sequence $\{f_1, \ldots, f_T\}$ representing a single substep and an initial segmentation mask $M_1$, the goal is to predict masks $\{M_2, \ldots, M_T\}$ for subsequent frames where the part's identity remains constant (i.e., before it connects with other parts or becomes part of a new sub-assembly). The task evaluates models' ability to track parts despite occlusions, varying viewpoints, and similar-looking parts in real-world scenarios.

**Experimental Setup.** We evaluate SAM2 [43] and Cutie [48] on video segments corresponding to assembly substeps with at least 20 frames. We closely follow the experiment design of standard video object segmentation, ensuring a fair comparison with other benchmark datasets. For each test example, we initialize the mask of the target part with ground truth in the first frame of the substep. We evaluate the performance on subsequent frames. We report the standard J&F metric for video object segmentation [45].

**Results and Analysis.** Table 5 shows that both models perform worse on our dataset compared to existing benchmarks. SAM2's performance drops moderately, while Cutie shows a more significant decrease compared to existing benchmarks. The results highlight challenges presented by our dataset, including camera movements, the presence of parts with similar appearances and small parts, frequent occlusions, and extended assembly sequences.

## 5.5 Shape Assembly with Instruction Videos

Given a set of 3D parts $\{p_1, \ldots, p_N\}$ and an instruction video $\{f_1, \ldots, f_K\}$, the goal of this task is to predict the 6-DoF poses $\{\zeta_1, \ldots, \zeta_N\}$ for the 3D parts to assemble them into complete furniture. We decompose this problem into several sub-tasks, including key frame detection, assembled part recognition, pose estimation, and iterative assembly.

**Method.** We propose a modular video-based shape assembly pipeline that consists of the following steps: First, a keyframe detection model should be employed to identify the frames in which two parts or sub-assemblies are being combined. These frames typically provide a clear view of how the parts are being connected. Next, a segmentation and part identification model should be used to identify which 3D parts from the set of all 3D parts are being assembled in the frame and to determine their 2D locations in the image. Third, starting with the first keyframe, we estimate the poses of the parts being assembled and combine them into a sub-assembly $A_i$, which is a set of transformed parts $\{\zeta_i(p_i)\}_{i=1}^{K}$, where $K$ is the number of parts in the sub-assembly. When moving to the next keyframe, we estimate the pose of the sub-assembly $A_i$ from the previous keyframe and the part to

be assembled next. Incrementally transforming and combining the parts based on the estimated poses from the keyframes, we gradually build the whole furniture.

**Experimental Setup.** Since most components of the proposed modular approach have been tested individually in previous experiments, in this experiment, we evaluate how the accuracy of the keyframe detection method affects the final shape assembly results. We propose two experimental settings. In the first setting, we use the annotation in our dataset to extract frames where two parts are being connected, which are typically the last frame for each substep. We also use the annotated poses of the parts or sub-assemblies from our dataset instead of deploying any models to perform pose estimation. We aim to provide a reference for evaluating the performance of our shape assembly approach when accurate pose information is available. In the second setting, we detect keyframes from the videos using the GPT-4o [49] vision and language model. In this setting, we first sample frames from the video at a rate of one frame per second. Then, we ask GPT-4o to describe the scene in each sampled frame. Finally, we feed those descriptions back to GPT-4o and require the model to predict which frames are keyframes. This setting explores using a pretrained vision and language model for identifying assembly steps in the instruction video. We evaluate the accuracy of the final assembly by computing the Chamfer Distance between the assembled furniture and the groundtruth furniture.

**Results and Analysis.** Fig. 7 presents qualitative results for the shape assembly task in the two experimental settings. The method achieves a Chamfer Distance of 0.33 in the first setting. The inaccuracy in this setting can be attributed to cases in which object parts are not fully connected in the last frames of the substeps. In the second setting, we test on 15 videos and obtain a Chamfer Distance of 0.55. We observe that GPT-4o fails to identify the final assembly step in 5 of the 15 videos, resulting in incomplete furniture. In summary, we introduce a novel approach for shape assembly by leveraging instruction videos, demonstrating the effectiveness of integrating video-based guidance in the assembly process. However, our results suggest tremendous challenges in grounding instructional assembly videos and highlight potential areas for improvement.

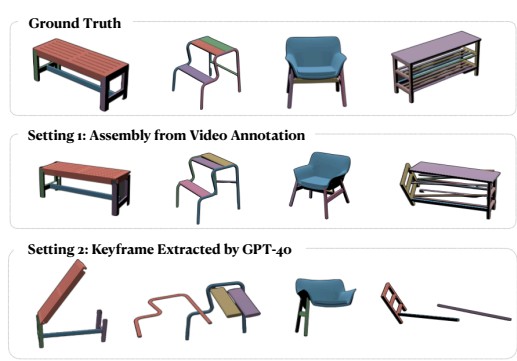

Figure 7: **Qualitative Examples.** Examples of shape assembly on the IKEA Video Manuals dataset.

## 6   Conclusion

In this paper, we introduce the IKEA Video Manuals dataset, a large-scale multimodal dataset with high-quality, spatial-temporal alignments of step-by-step instructions, 3D furniture models, and real-world assembly videos from the Internet. In total, our dataset provides 34,441 video frames annotated with part segmentations and 6-DoF poses for 98 assembly videos and 36 different IKEA furniture models from 6 furniture categories. Experiments on the dataset highlight significant challenges in grounding instructional assembly videos, including extracting part segmentations and poses, constructing high-level assembly plans, and detecting key assembly steps in videos.

**Limitations.** Currently, the dataset's limited size prevents large-scale training. The dataset focuses on visual and 3D information; including other data modalities such as audio or textual data is an important future direction. While manual annotation currently limits dataset scale, our framework for aligning Internet videos with 3D models and instructions provides a foundation for future expansion. The data collection process still requires manual annotation and verification, therefore presenting challenges for collecting data at a significantly larger scale. Current baselines demonstrate challenges in grounding 4D assembly but are yet to utilize this dataset for advanced model development. Since the dataset is focused on furniture assembly, whether models developed for this domain transfer to other assembly domains is left to be investigated. Future work could augment the dataset with additional modalities and develop algorithms leveraging instructional videos for 3D-grounded assembly plans. Broader implications include potential assistive technologies for individuals with disabilities, while internet-sourced data necessitates robust privacy and fair use methods.

## Acknowledgments and Disclosure of Funding

This work was in part supported by J.P. Morgan, the Stanford Center for Integrated Facility Engineering (CIFE), the Stanford Institute for Human-Centered Artificial Intelligence (HAI), NSF CCRI #2120095, RI #2211258, RI #2338203, ONR MURI N00014-22-1-2740, ONR YIP N00014-24-1-2117, and Microsoft. We extend our gratitude to Ruocheng Wang, Yunzhi Zhang, the members of the Stanford Vision and Learning Lab, and the anonymous reviewers for insightful discussions. We thank Yang Zhou for providing feedback on the paper. This paper was prepared for informational purposes in part by the CDAO group of JPMorgan Chase & Co and its affiliates ("J.P. Morgan") and is not a product of the Research Department of J.P. Morgan. J.P. Morgan makes no representation and warranty whatsoever and disclaims all liability, for the completeness, accuracy or reliability of the information contained herein. This document is not intended as investment research or investment advice, or a recommendation, offer or solicitation for the purchase or sale of any security, financial instrument, financial product or service, or to be used in any way for evaluating the merits of participating in any transaction, and shall not constitute a solicitation under any jurisdiction or to any person, if such solicitation under such jurisdiction or to such person would be unlawful.

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

# Supplementary Material for IKEA Manuals at Work: 4D Grounding of Assembly Instructions on Internet Videos

## Contents

## A   Dataset Details

IKEA Video Manuals is a multimodal dataset with high-quality, spatial-temporal alignments of step-by-step instructions, 3D object representations, and real-world video demonstrations from the Internet. IKEA Video Manuals provides 34,441 annotated video frames, aligning 36 IKEA manuals with 98 assembly videos for six furniture categories. Fig. A1 shows all 3D furniture models included in the dataset. An example of the annotations associated with each frame is shown in Fig. A2. We provide details of the data and annotations associated with each frame below.

**Furniture-level information**

- **Category:** The category label of the furniture (e.g., Bench).
- **Name:** The furniture name (e.g., applaro).
- **Furniture IDs:** A list of IKEA product IDs for the furniture.
- **Variants:** A list of furniture variants, if applicable.
- **Furniture URLs:** A list of IKEA product page URLs for the furniture.
- **Furniture Main Image URLs:** A list of URLs for the main product images on the IKEA website.

**Video-level information**

- **Video URL:** The URL of the video.
- **Additional Video URLs:** A list of additional video URLs for the same furniture.
- **Title:** The title of the video.
- **Duration:** The duration of the video (in seconds).
- **Resolution:** The resolution of the video (e.g., 1920x1080).
- **FPS:** The frame rate of the video (e.g., 30).
- **People Count:** The number of people in the video.
- **Person View:** The view of the person in the video (e.g., front, side).
- **Camera Fixation:** The fixation of the camera in the video (e.g., static, moving).
- **Indoor/Outdoor Setting:** The setting of the video (e.g., indoor, outdoor).

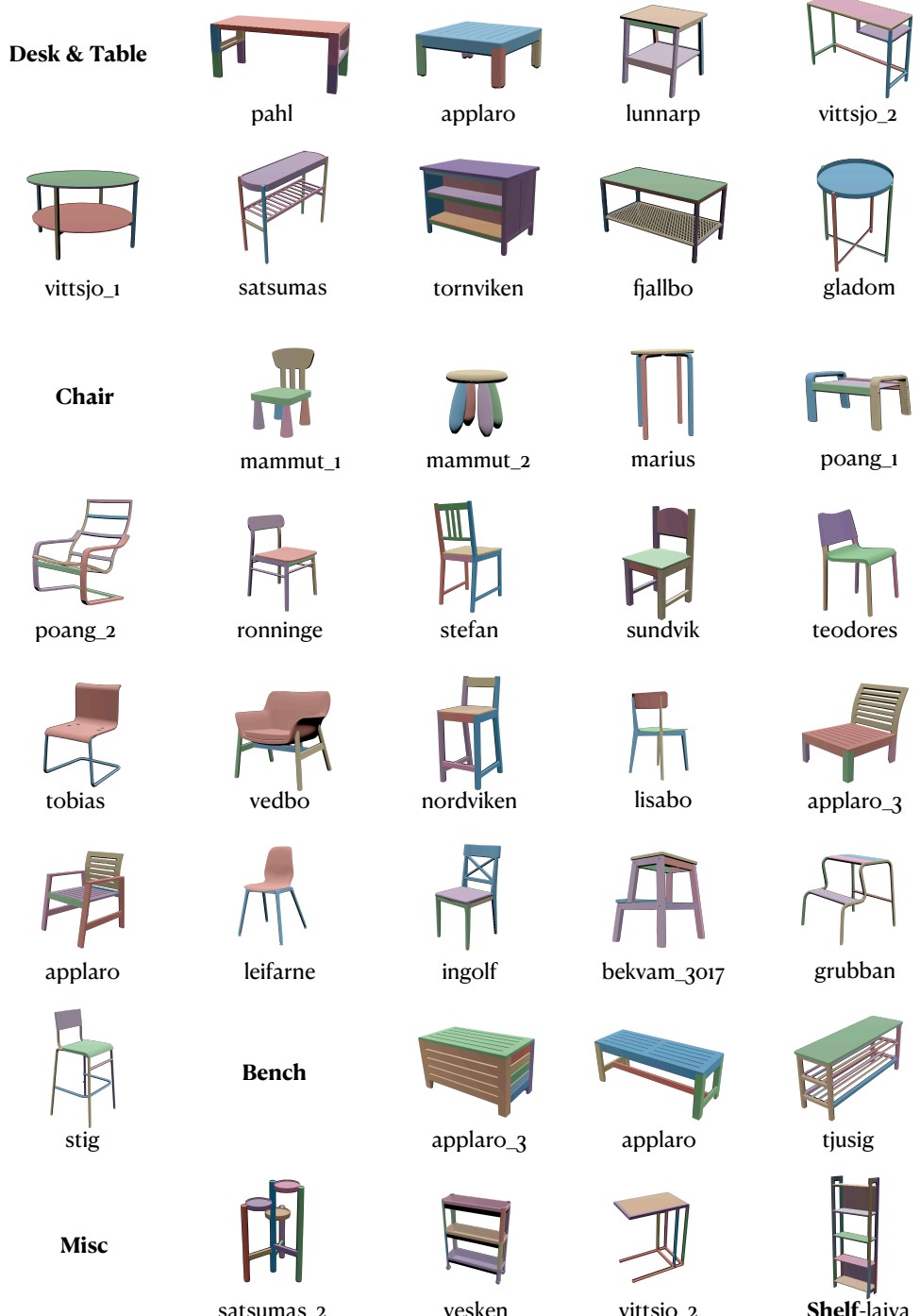

Figure A1: All furniture items included in the IKEA Video Manuals dataset, categorized by type–Desk, Table, Chair, Bench, and Misc.

**Assembly step information**

- **Step ID:** An unique ID is assigned to the assembly step.
- **Step Start:** The start time of the assembly step is shown in the video.
- **Step End:** The end time of the assembly step is shown in the video.
- **Substep ID:** The unique ID assigned to the substep within the assembly step.
- **Substep Start:** The start time of the substep in the video.
- **Substep End:** The end time of the substep in the video.

**Frame-level information**

- **Frame Time:** The timestamp of the frame in the video.
- **Number of Camera Changes:** The number of camera changes that have been labeled before the current frame.
- **Frame Parts:** A list of parts that are labeled in the frame (e.g., $[\{0, 2\}, \{1\}, \{3\}]$). The sub-assemblies that have been constructed in previous steps are denoted by a tuple of the part IDs.
- **Frame ID:** An unique identifier for the frame (e.g., 1584).
- **Is Keyframe:** A boolean value indicating whether the frame is a keyframe.
- **Is Frame Before Keyframe:** A boolean value indicating whether the frame is immediately before a keyframe.
- **Frame Image:** The RGB image of the frame.
- **Annotated Masks:** A list of segmentation masks for the parts in the frame.
- **Annotated Poses:** A list of 6D poses for the parts in the frame.

**Manual information**

- **Manual Step ID:** A unique ID assigned to the assembly step. This ID is associated with the step IDs in frame annotations.
- **Manual URLs:** A list of URLs for the assembly manual PDFs.
- **Manual ID:** An unique ID of the assembly manual from IKEA.
- **Manual Parts:** A list of part IDs is shown in the corresponding manual step.
- **Manual Connections:** List of connections between parts in the manual step.
- **PDF Page:** The page number of the manual step in the PDF.
- **Cropped Manual Image:** The cropped image of the corresponding manual step.

## B  Datasheets

**Dataset description.** The datasheet for the IKEA Video Manuals dataset, available at `https://github.com/yunongLiu1/IKEA-Manuals-at-Work/blob/main/datasheet.md`, including key aspects of data collection and annotation:

- **Consent:** The dataset is built upon two existing datasets, IKEA-Manual and IAW, which are publicly available for research purposes under the Creative Commons Attribution 4.0 International (CC-BY-4.0) license.
- **Personally Identifiable Information and Offensive Content:** The dataset does not contain any personally identifiable information or offensive content, as it focuses on furniture objects and assembly instructions.
- **Annotation Process and Compensation:** The data annotation process was outsourced to an annotation company. The annotators were compensated based on the work they provided, with the estimated hourly pay being above the minimum wage.

Please refer to the datasheet for more detailed information on the dataset.

**Link and license.** The dataset is available for public access under the CC-BY-4.0 license: `https://github.com/yunongLiu1/IKEA-Manuals-at-Work`

**Maintenance.** The dataset is hosted on GitHub and will be maintained by the authors. The repository can be found at: `https://github.com/yunongLiu1/IKEA-Manuals-at-Work`. The dataset has the following DOI: `https://doi.org/10.5281/zenodo.11623997`

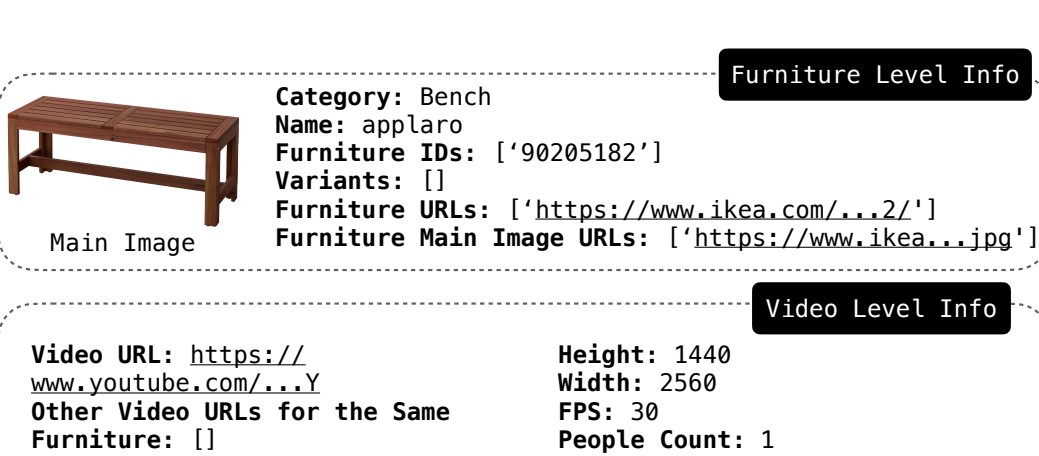

**Furniture Level Info**

**Category:** Bench
**Name:** applaro
**Furniture IDs:** ['90205182']
**Variants:** []
**Furniture URLs:** ['https://www.ikea.com/...2/']
**Furniture Main Image URLs:** ['https://www.ikea...jpg']

Main Image

**Video Level Info**

**Video URL:** https://www.youtube.com/...Y
**Other Video URLs for the Same Furniture:** []
**Title:** IKEA assembly instructions, ÄPPLARÖ Bench
**Duration:** 155

**Height:** 1440
**Width:** 2560
**FPS:** 30
**People Count:** 1
**Person View:** firstPerson
**Is_fixed:** fixed
**Is_indoor:** indoor

**Assembly Step Info**

**Step ID:** 1
**Step Start:** 47.0
**Step End:** 62.1

**Substep ID:** 3
**Substep Start:** 50.82
**Substep End:** 57.02

**Frame Level Info**

**Frame Time:** 52.82
**Number of Camera Changes:** 1
**Frame Parts:** ['0,2', '1', '3']

**Frame ID:** 1584
**Is Keyframe:** False
**Is Frame Before Keyframe:** False

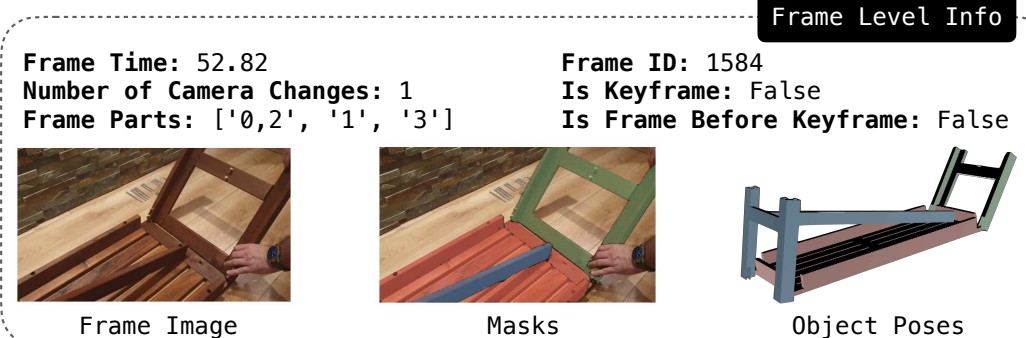

Frame Image                Masks                Object Poses

**Manual Level Info**

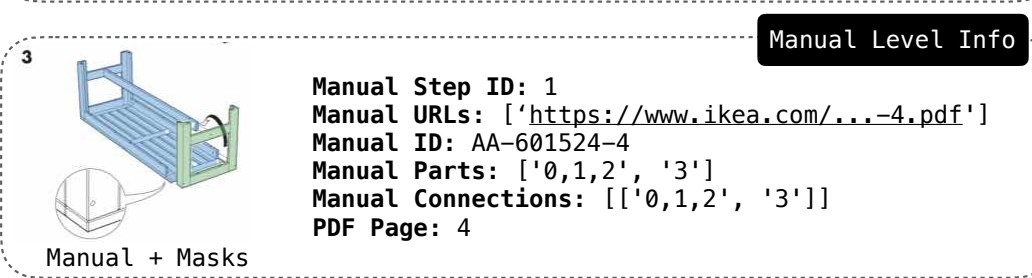

**Manual Step ID:** 1
**Manual URLs:** ['https://www.ikea.com/...−4.pdf']
**Manual ID:** AA−601524−4
**Manual Parts:** ['0,1,2', '3']
**Manual Connections:** [['0,1,2', '3']]
**PDF Page:** 4

Manual + Masks

Figure A2: An example of an annotated frame in the IKEA Video Manuals dataset. The annotation and data are divided into furniture-level, video-level, assembly step-level, frame-level, and manual-level information.

**Author statement.** The authors bear all responsibility in case of violation of rights. All annotations were collected by the authors, and we are releasing the dataset under CC-BY-4.0.

**Format.** The dataset contains videos, 3D models, manual PDFs, and annotated data (including temporal step alignments, temporal substep alignments, 2D-3D part correspondences, part segmentations, part 6D poses, and estimated camera parameters). The annotated data is stored in the JSON format. Other data are stored in their original formats and uploaded in a zip file. Upon decompression, the dataset is organized into subdirectories for videos, 3D models, and manual PDFs. Each is organized into subdirectories for furniture categories and further subdirectories for individual furniture items.

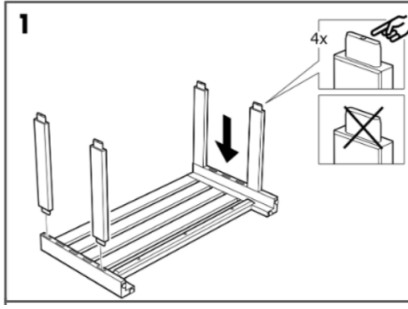

Figure A3: An example of an assembly step from the IKEA instruction manual that involves the assembly of four parts.

**Croissant Metadata** We will provide the structured metadata (schema.org standards) in `https://github.com/yunongLiu1/IKEA-Manuals-at-Work/metadata.json`.

## C Details for Data Annotation

To create the IKEA Video Manuals dataset, we identified 36 IKEA objects from the IKEA-Manual dataset [2] that have corresponding assembly videos in the IAW dataset [5]. We matched the unique IDs of the instruction manuals to ensure correct correspondence between the datasets. We provide additional details for each of the annotation steps below.

### C.1 Annotating Assembly Steps

For each assembly step annotated in the IKEA-Manual dataset [2], we identify matching video segments in the IAW dataset [5]. We manually adjust the start and end time of each video segment to include a more complete assembly process, from picking up a part to positioning and tightening. The adjustment ensures better alignment with the physical assembly actions.

### C.2 Annotating Assembly Substeps

In the IKEA instruction manuals, each single step may involve the assembly of multiple parts (as shown in Fig. A3). We provide a more fine-grained assembly process by introducing *substeps*. A substep is labeled when 1) a new part appears in the video or 2) a new sub-assembly is created through positioning and/or fastening of parts. On average, each assembly step contains 7.59 substeps. In total, the IKEA Video Manuals dataset contains 1120 substeps.

### C.3 Annotating Part Identities

In our dataset, each part of the 3D furniture model is assigned a unique ID consistent with the IKEA-Manual dataset. However, locating an individual 3D furniture part in the frame can be challenging due to several ambiguities, as illustrated in Fig. A4:

(a) Wrongly assembled parts that are initially placed incorrectly and later relocated, causing confusion for the annotator (Fig. A4a).

(b) Similar or identical-looking parts, such as chair legs, that are difficult to distinguish and label accurately (Fig. A4b).

(c) Parts that are heavily occluded, making it challenging to recognize the parts and their boundaries (Fig. A4c-d).

To ensure accurate part tracking throughout the video, we manually label the parts in the first frame of each substep after watching the entire video. This annotation is crucial for maintaining consistency when annotators only see individual frames in subsequent annotations. This approach ensures consistent part identities throughout the video, addressing challenges posed by heavy occlusions, similar-looking parts, and assembly mistakes.

### C.4 Annotating Segmentation Mask

To efficiently generate segmentation masks for furniture parts in video frames, we utilize the Segment Anything Model (SAM). When SAM fails to generate accurate masks (e.g., Fig. A5), we use a brush tool built into our annotation interface to refine the masks manually.

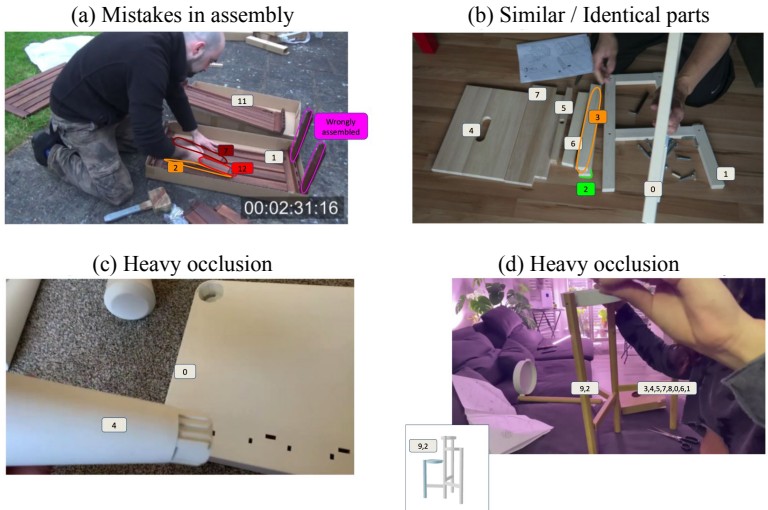

Figure A4: Examples of ambiguities in annotating part identities in assembly videos: (a) wrongly assembled parts later relocated, (b) similar-looking parts that are difficult to distinguish, (c-d) heavily occluded parts and boundaries between parts.

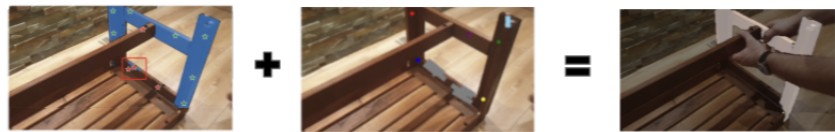

Figure A5: An example of refining the part segmentation using the brush tool. The initial segmentation is generated by the Segment Anything Model (SAM) model.

## C.5 Annotating 2D-3D Keypoints

To establish correspondence between the 3D parts and their 2D projections in the video frames, we annotate keypoints on both the 3D parts and the 2D images. Our annotation interface is shown in Fig. A6. The annotation interface computes the part poses and camera parameters using the Perspective-n-Point (PnP) algorithm and visualizes the 2D projection in real-time. Based on the visualization, the annotator can interactively refine the keypoint annotations to maximize the overlap between the 2D projection and the part seen in the 2D image.

## C.6 Robust Camera Parameter Estimation

A key challenge in annotating real-world assembly videos is accurately estimating camera parameters, which can vary due to factors such as focal length adjustments and switching between multiple cameras. Our data collection pipeline incorporates several steps to address these issues:

**Camera Change Detection:** We manually annotate points in the video where camera changes occur. This allows us to segment the video into regions with consistent camera parameters.

**Per-Segment Intrinsic Estimation:** For each video segment between camera changes, we estimate a set of intrinsic camera parameters. This approach allows for consistent parameters within a segment while accommodating changes between segments.

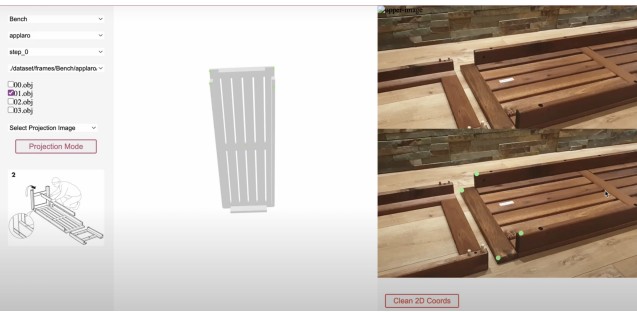

Figure A6: The annotation interface for labelling keypoint correspondences between 3D models and 2D video frames.

**Keypoint-Based PnP and RANSAC:** We use manually annotated 2D-3D keypoint correspondences and apply the Perspective-n-Point (PnP) algorithm, combined with RANSAC for robustness, to estimate camera pose and refine intrinsic parameters.

**Multi-Candidate Optimization:** We generate multiple candidate intrinsics for each segment and select the one that minimizes reprojection errors across the segment.

**Cross-Frame Consistency:** We enforce temporal consistency by initializing pose estimates in each frame based on the previous frame's refined poses.

This multi-step approach allows us to handle the complexities of real-world videos, including varying camera parameters and viewpoints, ensuring accurate and consistent 3D annotations throughout the assembly process. This robustness is crucial for the reliability of our dataset and distinguishes it from approaches that assume constant camera parameters or controlled environments.

### C.7 Part Identification and Annotation Consistency

To ensure accurate part identification, we assign unique IDs to 3D model parts, matching the IKEA-Manual dataset. Annotators watch the entire video before starting, determining part IDs based on 3D model correspondence. For similar parts (e.g., table legs), we reference IKEA-Manual's assembly order. If leg_3 is first in IKEA-Manual, we label the first assembled leg as 3 in our video, with other similar parts identified by their relative positions.

To maintain annotation accuracy and consistency, our interface requires annotators to focus on one part throughout the video before moving to the next. We conduct multiple rounds of checks and re-annotations, with pose annotations beginning only after the mask is verified. Poses are initialized from previous frames to maintain consistency. This comprehensive approach ensures reliable part tracking, even for visually similar components, throughout the assembly process.

In our dataset, we create new substeps whenever a new part appears or a sub-assembly is formed. This approach ensures all relevant parts or sub-assemblies are visible in the first frame of each substep. If a part and all its connected components (i.e., the whole sub-assembly) disappear after being visible in the first frame, it will still be kept in the assembly process. However, we mark its mask and pose as "not visible." There are no instances in our dataset where a part remains unclear throughout an entire substep.

### C.8 Details on locating 3D parts

We first segment each video into substeps when a new part appears or a new subassembly is formed. For the first frame of each substep, we annotate part identities that are consistent throughout the video. The annotators then annotate masks for each part across all frames, which are verified for consistency manually. After mask verification, we proceed to 3D pose annotation with a custom interface that allowed annotators to view the parts from multiple angles to ensure accurate relative poses. We pay particular attention to coplanarity, inter-part distances, and correct relative locations (right/left, front/back, up/down). This multi-step approach, detailed in Sections 4.2-4.4 of our paper, allows us to accurately locate and pose 3D furniture parts in real-world assembly videos.

### C.9 Pose Refinement

While initial estimates of the part poses can be obtained from the annotated 2D and 3D keypoints, these estimates are often inaccurate, particularly in terms of the relative positions and orientations of

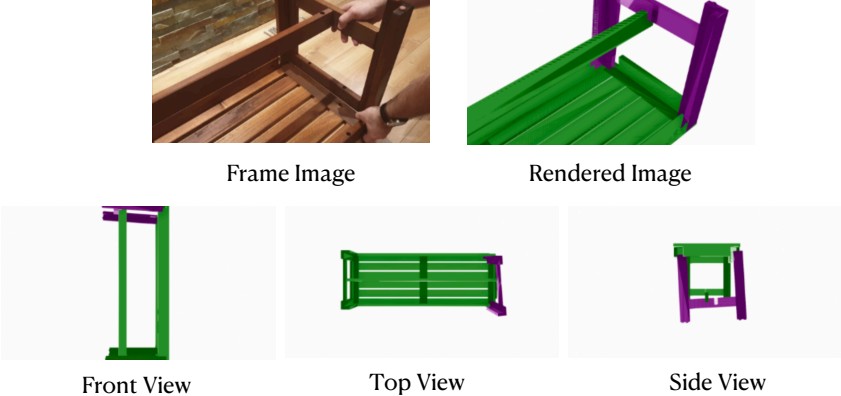

Figure A7: An example of an inaccurate part pose estimated from the 2D-3D keypoints. Despite that the 2D projections of the 3D models overlap with the parts in the video frame (top row), the 3D poses of the parts can be found incorrect when visualized in 3D and viewed from other angles (bottom row).

the parts in 3D space. Fig. A7 shows an example where the 2D projection of a part appears correct, but when viewed in 3D, the part is positioned incorrectly relative to other parts. To address this issue, we developed an interface that allows annotators to refine the initial estimate by rotating and translating each part in 3D space. The annotators can view 3D parts from different viewpoints to ensure that the relative poses of the parts are correct and consistent with the assembly process seen in the videos.

We take an additional step in the pose refinement process to help maintain the temporal smoothness of the part trajectories. During the pose refinement process, the initial poses of the parts in a frame are set to the refined poses of the corresponding parts from the previous frame. This initialization strategy helps to reduce large changes in the annotated part poses between neighboring frames, resulting in more coherent and realistic pose trajectories.

# D    Details for Quality Control

To ensure the accuracy and consistency of the annotations in the IKEA Video Manuals dataset, we analyze the common errors in the annotations and perform extensive verifications.

## D.1    Common Errors

For mask annotations, common errors include incorrect part segmentation, missing parts, and noisy masks. Incorrect part segmentation occurs when annotators misidentify the boundaries of a part due to similar colours or complex shapes. Missing parts occur when certain parts are not segmented, often due to occlusions. Noisy masks often occur when the SAM model fails to generate accurate masks, leading to incomplete or inaccurate segmentation.

For pose annotations, common errors include incorrect part identification, incorrect relative poses, and interpenetrations. Incorrect part identification occurs when the annotators annotate an incorrect part, leading to an incorrect pose. Incorrect relative poses occur when the estimated pose does not accurately reflect the actual position and orientation of the part relative to other parts in 3D space. Interpenetrations occur when parts intersect or overlap in 3D space, leading to unrealistic poses.

## D.2    Extensive Verification

We conduct extensive verification to ensure the high quality of the mask and pose annotations. The verification interface (as shown in Fig. A8) displays the original video frame, the video frame overlaid with segmentation masks, the video frame overlaid with 2D projections based on estimated poses, and the 3D parts in the estimated poses from different viewpoints. In particular, for mask annotations, we verify if the 2D mask corresponds to the correct part, covers the entire part, does not contain pixels of other parts, and is free of noise due to limitations of the Segment Anything Model (SAM). For pose annotations, we verify if the pose annotation corresponds to the correct part, the 2D projection aligns with the part in the frame image, and the 3D parts have correct relative poses. We automatically filter out pose annotations with interpenetrations between parts.

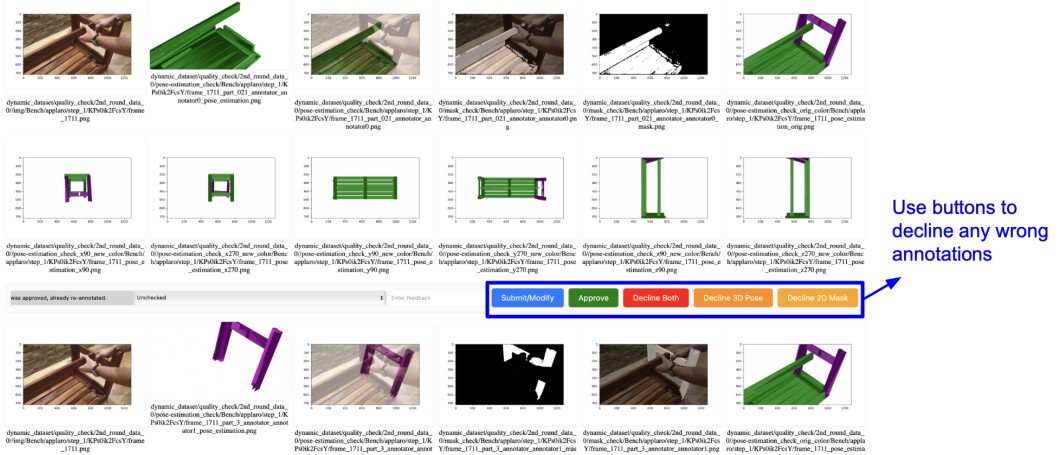

Figure A8: The verification interface for assessing the quality of mask and pose annotations. The interface visualizes the original video frame, mask overlay, pose overlay, and 3D parts in estimated poses from different viewpoints.

# E    Annotation Interfaces and Instructions

This section provides details of the instructions given to annotators. Instructions for using these interfaces are mainly provided through demonstration videos, which are included in the project's GitHub repository (https://github.com/yunongLiu1/IKEA-Manuals-at-Work) for reference.

## E.1    Segmentation Mask and 2D-3D Points Correspondence Annotation Interface (Fig. A6)

The Segmentation Mask and 2D-3D Points Correspondence Annotation Interface allow annotators to generate segmentation masks and establish correspondences between 3D models and 2D video frames. Annotators can switch between the two annotation modes using a dedicated button in the interface. The following steps outline the annotation process:

1. Select the appropriate category, subcategory, object, and step for the video you want to annotate.

2. In the Segmentation Mask mode:
    - Select points that best represent the overall shape and area of the part to ensure optimal performance of the Segment Anything Model (SAM).
    - Use the provided tools, such as a brush or eraser, to refine the mask based on the feedback provided.

3. In the 2D-3D Points Correspondence mode:
    - Select corresponding points on the 3D model and the 2D video frame that represent key features or edges of the furniture parts.
    - Review the rendered image and adjust the selected points if necessary to improve the alignment between the 3D model and the 2D video frame.

4. Navigate frames using the 'Next Frame' button and review the predicted points from the TAPIR model, modifying any unsatisfactory points.

5. Review the segmented images and estimated poses for accuracy and consistency, and submit the annotations.

## E.2    Mask Re-Annotation Interface (Fig. A9)

1. **Review Previous Mask:**
    - The interface will display the previously annotated mask in the bottom left corner of the screen, along with the reason for the decline. Reviewing the previous mask and the reason for the decline helps the annotator understand the required corrections.

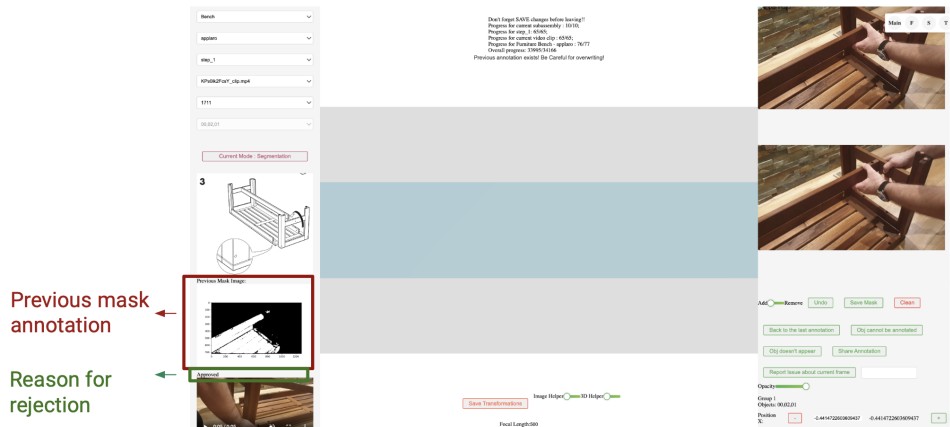

Figure A9: The interface for correcting and refining mask annotations based on feedback. The interface provides tools for manual refinement of the segmentation masks, including a brush and an eraser.

- Reasons for the decline include "mask was annotated to the wrong part," "mask did not include the whole part/include other parts," or "noisy mask, which is normally caused by the limitation of SAM and can be solved by using the brush." These specific reasons guide the annotator in refining the mask.

2. **Refine Mask:**
   - Use the provided tools, such as a brush or eraser, to refine the mask based on the feedback provided. These tools allow precise modifications to the mask.
   - Ensure the refined mask accurately captures the entire part while excluding any neighbouring parts or background. An accurate and complete mask is essential for downstream tasks.
   - Pay attention to the edges and boundaries of the part to create a clean and precise mask. Well-defined edges improve the quality and usability of the mask.

3. **Additional Buttons:** (Same as in the Segmentation Mask Annotation Interface.)

4. **Review and Submit:** Review the refined mask for accuracy and completeness, ensuring it addresses the reason for the decline. This final review step verifies that the necessary corrections have been made. Submit the updated mask using the provided submission functionality to save the work and proceed to the next task.

### E.3 Pose Refinement Interface (Fig. A10)

The Pose Refinement Interface enables annotators to refine the initial poses estimated from the previous annotation. The following steps outline the pose refinement process:

1. Review the initial poses of all parts in the frame, estimated from the previous annotation.

2. Use the provided controls to adjust the position and orientation of each part in the camera frame.

3. Ensure that the relative positions and orientations of the parts are consistent with the assembly process.

4. Review the refined poses for accuracy and submit the updated poses.

By following these instructions and leveraging the provided video demonstrations, annotators can effectively use the annotation interfaces to generate high-quality segmentation masks, 2D-3D point correspondences, and refined part poses for the IKEA Video Manuals dataset.

## F  Error Bar

To assess the variability of our model's performance, we run experiments on a subset of the data with 3 different random seeds for both the segmentation and pose estimation tasks. For part-conditioned

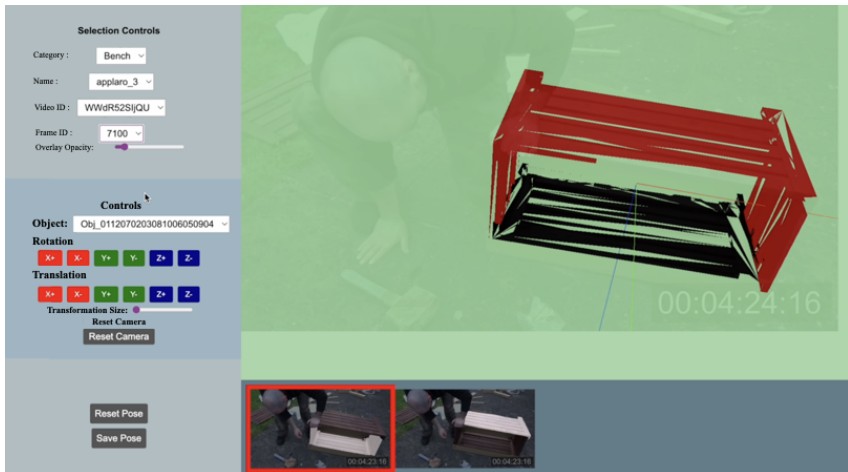

Figure A10: The interface for manually adjusting part poses in 3D. The interface supports adjusting the 3D position and orientation of each part. The interface also provides visualization of the 3D parts from different perspectives.

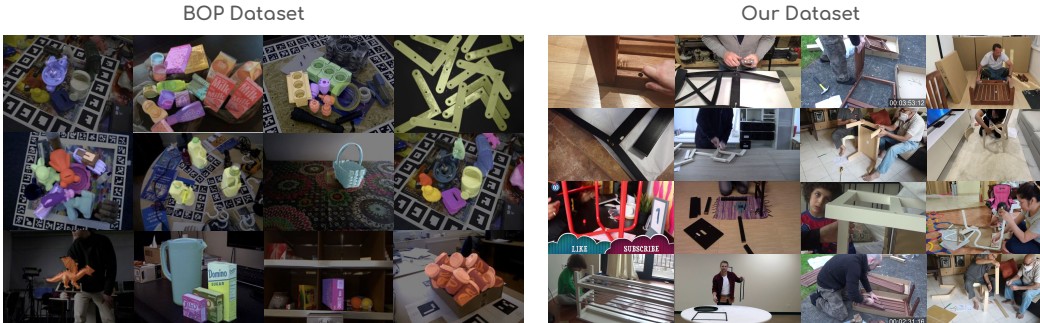

Figure A11: Overview of BOP dataset (left) and IKEA Video Manuals dataset (right).

segmentation, the standard deviation of the IoU metric is 0.01 for the CNOS method and 0.03 for the SAM-6D method. When considering the Top-5 IoU, the standard deviations are 0.08 and 0.09 respectively. For part-conditioned 6D pose estimation, the SAM-6D method has a standard deviation of 0.12 for the ADD score and 0.08 for the ADD-S score. The MegaPose method has standard deviations of 0.09 and 0.05 for ADD and ADD-S. These results indicate that the performance of these models on our dataset remains relatively consistent overall.

## G  Compute Resources

The computational resources used for this project were computing nodes from the Stanford SC computational cluster. We used around 40 jobs lasting 7 days for running segmentation experiments using SAM-6D and CNOS, and pose estimation experiments using SAM-6D and MegaPose. The jobs were assigned to nodes equipped with different NVIDIA GPU models, including 2080 Ti, Titan RTX, 3090, A40, A5000, A6000, and L40S, based on availability.

## H  Error Analysis for 6D Pose Estimation Models

This section presents an error analysis of SAM-6D and MegaPose on the IKEA Video Manuals dataset, aiming to identify specific scenarios where these models fail. By examining these failure cases, we provide insights for future improvements in 6D pose estimation algorithms, particularly for real-world videos.

### H.1  Dataset Overview

Fig. A11 provides an overview of the BOP dataset and our IKEA Video Manuals dataset. While BOP features small objects in controlled environments, our dataset captures furniture in diverse,

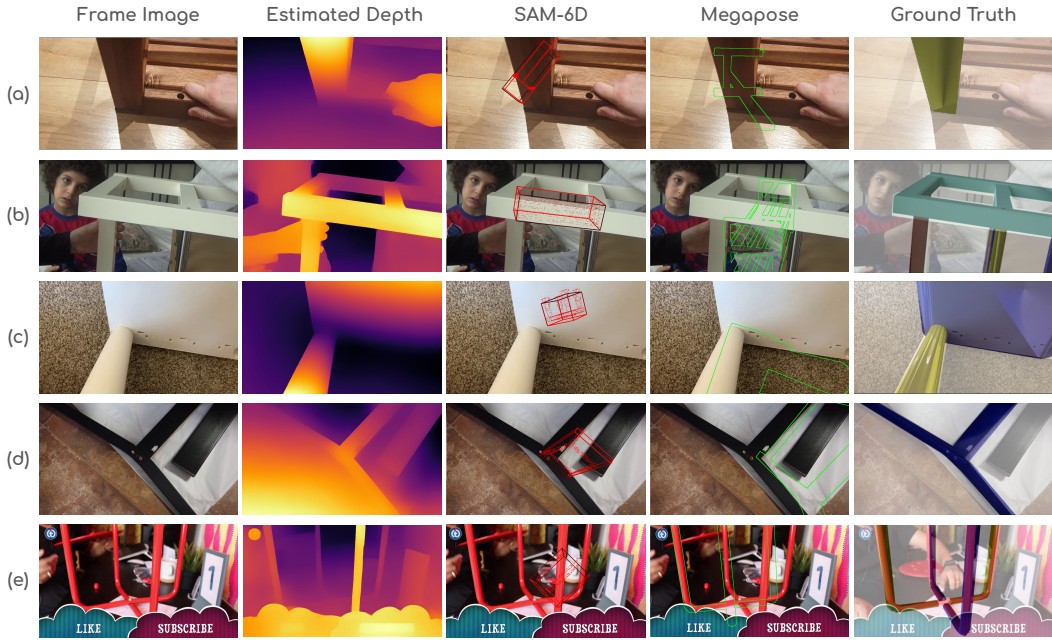

Figure A12: Failure cases: Close-up view with partial visibility.

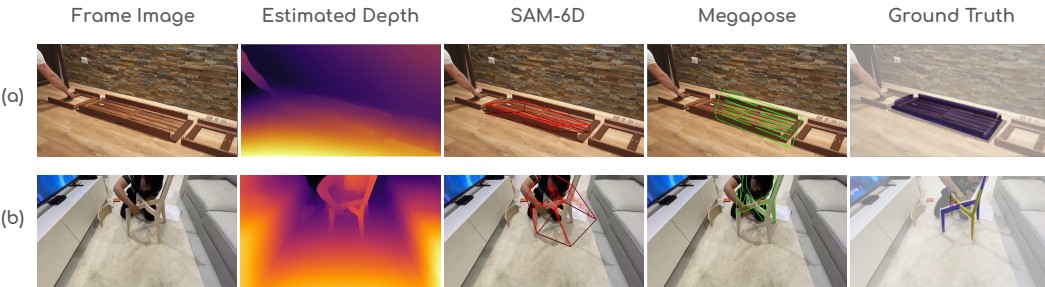

Figure A13: Failure cases: Ambiguous semantic information leading to orientation uncertainty.

real-world settings. This difference in complexity and scene composition presents more challenging scenarios for pose estimation algorithms, contributing to the performance degradation observed in our experiments.

## H.2 Analysis of Failure Cases

We identify four primary categories of scenarios that lead to failures in both SAM-6D and MegaPose:

### H.2.1 Close-up Views with Partial Visibility

Fig. A12 illustrates a common scenario in our dataset where only a small portion of the furniture is visible due to close-up views. Both SAM-6D and MegaPose struggle to infer the complete object pose from this limited visual information. Both of SAM-6D and MegaPose fail to accurately determine the scale of the object. This scenario, while frequent in real-world furniture assembly, is rare in existing datasets, highlighting a key area for improvement.

### H.2.2 Ambiguous Semantic Information

Fig. A13 demonstrates the challenge posed by objects with similar appearances from different angles. In example Fig. A13(a), we see a furniture piece with similar front and back views. MegaPose, which relies heavily on visual features, incorrectly estimates the orientation by 180 degrees. SAM-6D performs slightly better due to its incorporation of depth information, but still shows uncertainty in its pose estimate, as evidenced by the misaligned bounding box.

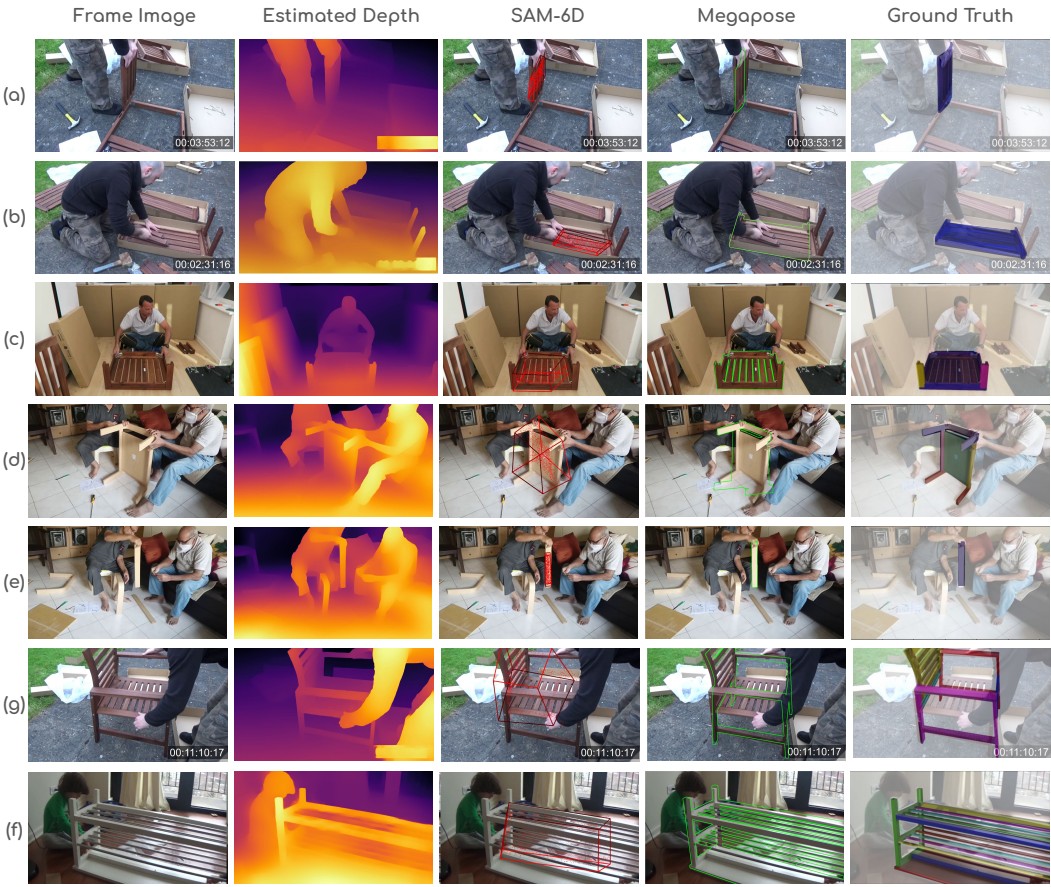

Figure A14: Failure cases: Depth discontinuities due to occlusions.

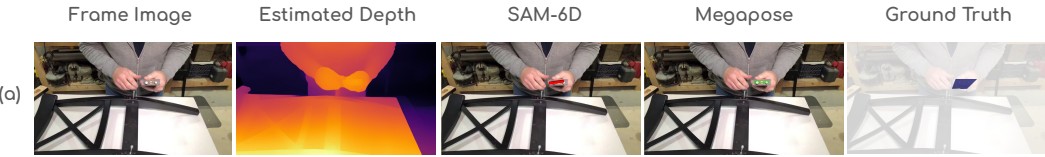

Figure A15: Failure cases: Challenging viewpoints leading to scale and shape misinterpretation.

### H.2.3 Depth Discontinuities Due to Occlusions

Fig. A14 showcases how occlusions, such as hands in front of the target object, introduce depth discontinuities that particularly affect depth-based models. SAM-6D struggles to differentiate between the depth of the occluder and the target object, leading to an inaccurate pose estimate where the predicted bounding box excludes the occluded region. While MegaPose is less affected due to its appearance-based approach, it still shows reduced accuracy in estimating the object's orientation under these conditions.

### H.2.4 Challenging Viewpoints

Our dataset includes a variety of viewpoints that, while common in real-world scenarios, pose significant challenges for pose estimation methods (Fig. A15). These include top-down views of furniture legs, partially obscured angles, and perspectives that make it difficult to discern the full scale or shape of the object. In these cases, both seen and unseen object methods often misinterpret the scale or shape of the object, leading to incorrect pose estimates. This scenario highlights the

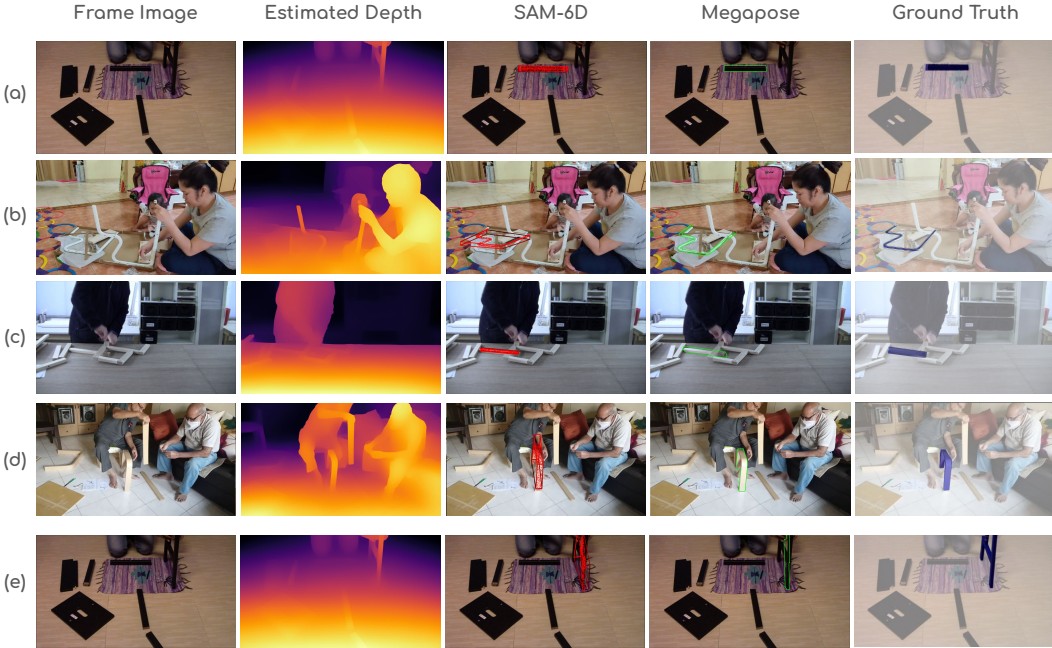

Figure A16: Success cases: Full object visibility with minimal occlusion.

importance of capturing and evaluating a diverse range of realistic viewpoints, which our dataset provides.

### H.2.5 Full Object Visibility with Minimal Occlusion

Our dataset also includes cases where the entire object is visible with minimal occlusion (Fig. A16). In these instances, both SAM-6D and MegaPose perform better compared to more challenging scenarios, though not achieving perfect accuracy. This case serves as a useful comparison point, demonstrating the relative improvement in performance under more favorable conditions and highlighting the impact of the more challenging cases on overall accuracy.

### H.3 Discussion

Our error analysis reveals that the IKEA Video Manuals dataset presents more challenging scenarios than typical object pose datasets, accurately reflecting the complexities of real-world furniture assembly tasks. The less-defined shapes of furniture compared to small, manufactured objects make silhouette-based approaches less reliable, while the wide range of real-world challenges—including partial visibility, occlusions, ambiguous views, and diverse viewpoints—test the limits of current algorithms.

While our analysis focuses on furniture assembly, many of these challenges have broader implications for 6D pose estimation in general. Issues such as partial visibility, ambiguous semantic information, and challenging viewpoints are encountered across various domains, from industrial assembly to robotic manipulation in unstructured environments. However, furniture assembly presents a unique combination of these challenges, often involving large, multi-part objects with complex spatial relationships, making it an excellent proxy for a wide range of real-world pose estimation problems.

## I  Dataset Comparison

This section provides a comprehensive comparison of our IKEA Video Manuals dataset with existing assembly datasets, expanding on the comparison presented in Table 1 .

Fig. A17 illustrates the diversity of object types in our dataset compared to existing assembly datasets. As shown, we cover 36 distinct furniture items with 268 individual parts across 6 categories, including chairs, tables, benches, desks, shelves, and other miscellaneous items. This diversity exceeds other

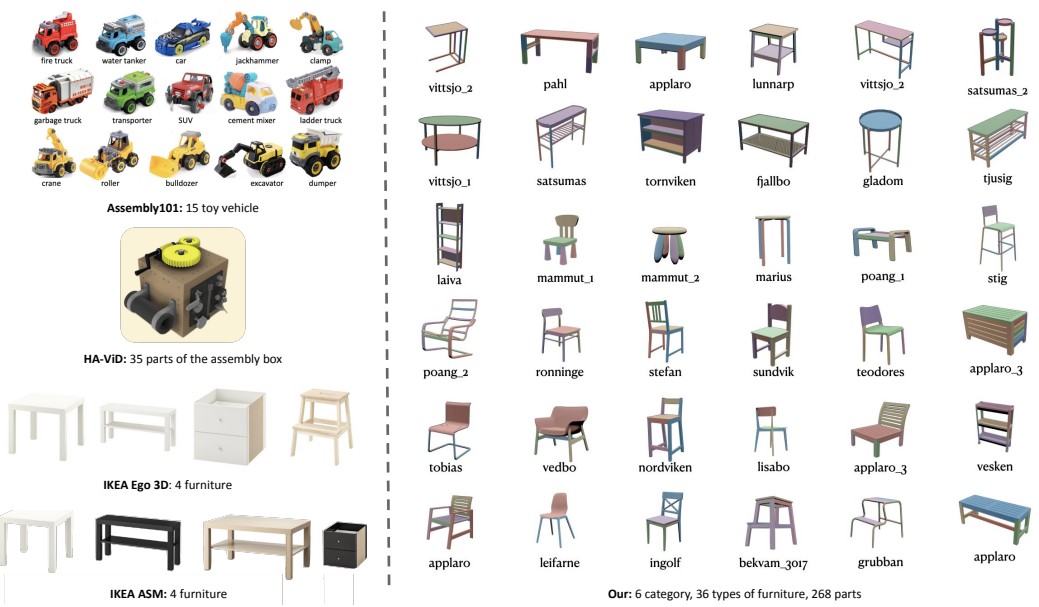

Figure A17: **Object Types Comparison.** Our dataset covers a diverse range of furniture types, significantly surpassing the variety found in existing furniture assembly datasets with 3D information.

Table A1: **Full comparison of IKEA Video Manuals with existing assembly datasets.** The table provides a comprehensive comparison, covering various aspects such as object category, video source, 3D info, and action labels.

| Dataset | # Object Category | # Objects | Object Type | Video Source | # Environments | Environment Type | # Assemblers |
|---|---|---|---|---|---|---|---|
| IKEA Video Manuals (Ours) | 6 | 36 | Furniture | Internet | ∼90 | Indoor/Outdoor, Fixed/Moving Camera, First-/Third-Person View | Multiple |
| Assembly101 [11] | 15 | 101 | Toy vehicles | Lab | 1 | Lab, 8 Fixed + 4 Egocentric Cameras | 53 adults |
| HA-ViD [30] | 1 | 35 parts | Generic assembly box parts | Lab | 1 | Lab, 3 Fixed Cameras | 30 participants |
| IKEA-Manual [2] | 6 | 102 | Furniture | - | - | - | - |
| IKEA Ego 3D [31] | 4 | 4 | Furniture | Lab | 1 | Lab, Egocentric | 2 |
| IKEA ASM [3] | 3 | 4 | Furniture | Lab | 5 | Lab and home environments | 48 human subjects |
| IKEA in the Wild [32] | 14 | 420 | Furniture | Internet | ∼1000 | Indoor/Outdoor, Fixed/Moving Camera, First-/Third-Person View | Multiple |

| Dataset | Image Modality | # Manuals | 3D Object Model | 2D Info | 3D Info | Object Tracking | Camera Information |
|---|---|---|---|---|---|---|---|
| IKEA Video Manuals (Ours) | RGB | 36 | ✓ | Segmentation | 6-DoF Pose | ✓ | Estimated, consistent throughout video segment |
| Assembly101 | RGB + Monochrome | 0 | × | × | Depth | × | Calibrated multi-view cameras |
| HA-ViD | RGB + Depth | 0 | ✓ | Bounding Box | Depth | ✓ | Calibrated multi-view cameras |
| IKEA-Manual | - | 102 | ✓ | Segmentation | 6-DoF Pose | - | Estimated camera, different between parts |
| IKEA Ego 3D | RGB + Depth | 0 | × | × | Depth | × | Calibrated egocentric camera |
| IKEA ASM | RGB + Depth | 0 | × | Both | Depth | ✓ | Calibrated multi-view cameras |
| IKEA in the Wild | RGB | 461 | × | × | × | × | Diverse, uncalibrated |

| Dataset | # Action Labels | Human/Hand Pose | Limiting Factors for Extension | Year |
|---|---|---|---|---|
| IKEA Video Manuals (Ours) | 137 manual steps, 1120 substeps | × | Annotations | 2024 |
| Assembly101 | 1380 (fine-grained), 202 (coarse) | 3D hand pose | Objects, Camera Rig Setup, Participants | 2022 |
| HA-ViD | 75 (primitive tasks), 219 (atomic actions) | 2D + 3D | Generic Assembly Box Design, Participants | 2023 |
| IKEA-Manual | 393 manual steps | × | 3D Models, Manual Creation, Annotations | 2022 |
| IKEA Ego 3D | 56 atomic action | × | Furniture, Participants | 2024 |
| IKEA ASM | 33 atomic actions | × | Furniture, Participants | 2023 |
| IKEA in the Wild | 15649 action labels (manual step) | × | Video Availability, Annotations | 2023 |

furniture assembly datasets, including IKEA ASM (4 furniture types across 3 categories) and Ego 3D (4 furniture types across 4 categories).

Table A1 provides a comprehensive comparison of our IKEA Video Manuals with existing assembly datasets, elaborating on the summary presented in the main text. The table is divided into three sections, detailing dataset characteristics, data modalities and annotations, and action labels with additional features. This expanded comparison highlights unique aspects of our dataset, such as the diversity of environment types (indoor/outdoor, fixed/moving camera, first-/third-person view), the consistency of camera information throughout video segments, and the fine-grained nature of our action labels (137 manual steps and 1120 substeps). It also contextualizes our dataset within the broader landscape of assembly datasets, showcasing its contributions in terms of real-world complexity, diverse assembly sequences, and rich spatio-temporal annotations.

# J    Comparison with IKEA-Manual

Our IKEA Video Manuals dataset differs from the IKEA-Manual dataset in several key aspects:

**Detailed 3D Motion Information:** We provide detailed 3D motion information throughout furniture assembly. Unlike IKEA-Manual, which provides static poses for each assembly step, our dataset captures the entire assembly process with dense temporal sampling.

**Diverse Real-World Environments:** We capture assembly in diverse real-world settings. Our dataset leverages internet videos to capture assembly processes in a wide variety of environments, including different lighting conditions, backgrounds, and camera viewpoints. This is in stark contrast to IKEA-Manual, which uses more controlled images.

**Video-Based Part Assembly Task:** We propose a distinct part assembly task based on video information, as detailed in Section 5.5. This task differs fundamentally from the assembly task in IKEA-Manual, which relies solely on 3D part information, ignoring information from the manual images. Our approach incorporates temporal information from videos.

**Consistency in Camera Information:** Our dataset ensures consistency in camera information throughout video segments. This addresses a key challenge in real-world assembly videos where camera parameters may change due to focal length adjustments or switching between multiple cameras. In contrast, IKEA-Manual estimated intrinsics separately for each part, which could lead to unrealistic relative poses in 3D space.

**Challenges in Real-World Visual Grounding:** Despite benchmarking the same part segmentation and part pose estimation tasks as IKEA-Manual, our experiment results reveal significant challenges of visual grounding on real-world images (see Appendix H) instead of on manual images that aim to clearly illustrate the assembly process.

By providing dense spatio-temporal annotations on Internet videos, our dataset offers opportunities for studying different aspects of part assembly, such as 3D object tracking in videos and spatio-temporal action localization in videos. These features, combined with our diverse set of real-world assembly sequences, position our dataset as a unique and valuable resource for advancing research in assembly understanding and related fields.

