# OpenReview forum: "IKEA Manuals at Work: 4D Grounding of Assembly Instructions on Internet Videos"
_NeurIPS.cc/2024/Datasets_and_Benchmarks_Track — NeurIPS 2024 Track Datasets and Benchmarks Poster_

### Official Review · Reviewer_hZeh · 2024-07-15

**Rating:** 6
**Confidence:** 4
**Clarity:** Yes. Writing is clear and figures, Su…

**Review:**

This work identifies the useful task of furniture assembly and well-motivates 4D grounding annotation. However, the resulting dataset is not substantially novel in collection process nor application, and the result is a small-scale benchmark from which little information is gleaned. Please see strengths and opportunities for improvement for detail.

**POST-REBUTTAL EDIT** Thanks to the authors for their thorough rebuttal and the other reviewers for their thoughtful comments. My main concern was the dataset collection method and final application was not novel nor would have high impact. The rebuttal emphasizes the distinction of their work from previous work IKEA-Manual; namely that it requires grounding in images from the real-world rather than a manual. In addition, the rebuttal presents additional baselines to give the dataset more context and provide more impact. Based on this and agreement with other positive reviewers I am happy to improve my score to borderline accept.

**Strengths:**

Furniture assembly is a challenging and useful task. Grounding instructions in 4D provides holistic annotations for this task which serve as a solid benchmark for part assembly.
- The paper well-motivates 4D grounding: prior work focuses on planning or registration of 3D models to assembly steps. In order to learn both planning, grounding and control; needed for assembly, motion is also needed. To this end, this work produces 3D part registration to assembly video in addition to instruction manuals.
- The dataset can be used for part assembly, and can specifically assess subtasks in assembly such as plan generation, part segmentation and pose estimation.
- Using furniture containing 3D models enables less expensive 4D annotation

**Additional Feedback:**

- L40 “of our work include” should be “of our work include:”
- Table 1 Caption: “Genofn” should likely be “Generation”

**Correctness:**

Yes. "Opportunities for Improvement" suggests more comprehensive benchmarking, but the dataset appear to be correct as-is.

**Documentation:**

Yes, this is well-documented in the Supplement.

**Limitations:**

Yes. Section 6 and the Supplement provide a good

**Opportunities For Improvement:**

Main contribution of the work is modest

-	The main contribution of the work is part segmentation and pose. Part segmentation is handled by human annotated keypoints to prompt SAM segmentation. Part pose estimation uses annotated keypoint 2D-3D correspondence followed by PnP. These annotation methods have been previously used [16 from the paper, Segment Anything, uses human annotation to prompt it; 30 from the paper uses 2D-3D correspondence and PnP], so their academic contribution instead must be either the integration with this task and/or the resulting dataset.
-	The integration of this annotation with assembly is useful for the reasons stated in “Strengths.” However, prior work IKEA-Manual [30 from the paper] is already able to evaluate part segmentation, part pose estimation and part assembly. Therefore, the contribution here is to apply this to video as opposed to simply instruction steps, which is modest.
-	Because this annotation requires human annotation, it is not scalable. As a result, the dataset is small scale (~100 videos) can only be used for evaluation. While it is great to benchmark subtasks in assembly, the paper struggles to draw insight from this evaluation beyond “these are challenging tasks.” I therefore argue the dataset in itself is not a strong enough contribution.
-	The dataset could be a more substantial contribution were it to be useful for training a model, or as a more comprehensive benchmark. For instance, if more than two methods were be evaluated in each task, maybe more interesting insights could be gained.

**Relation To Prior Work:**

Yes, the related work is well-written.

**Summary And Contributions:**

This paper approaches the task of furniture assembly. It presents a collection of 98 assembly videos across 6 furniture categories and uses human annotation to label object part segmentation and pose at 1fps, for a total of 34k video frames. It uses this dataset to evaluate two existing heuristic baselines for assembly plan generation, evaluate two existing pre-trained learned methods on part segmentation, evaluate two pre-trained learned methods on part pose estimation, and evaluate shape assembly by combining the aforementioned methods applied to keyframes, detected by an LMM. It shows the methods struggle on these challenging tasks.

---

> ### Author Rebuttal · Authors · 2024-08-17
>
> ### **Novelty in the collection process and the application.**
> The main contribution of our dataset is a "novel multimodal dataset that captures the complexity of real-world furniture assembly tasks", as discussed in the introduction. Our added comparison table below highlights that we are **the first dataset that provides 6D pose annotations on internet videos** for furniture assembly. Therefore, our data collection pipeline (Figure 3) addresses challenges specific to internet videos that are not typically encountered when collecting data in lab environments or for manual images, therefore has value for future works on grounding 4D information on internet video data.
>
> To contribute this dataset, our data collection pipeline addresses key challenges in annotating real-world videos to ensure correct relative poses between 3D furniture parts and cross-frame consistency, as discussed in Section 4.4. In particular, we introduce steps to accurately estimate camera parameters from the videos. This is important because focal length adjustments or switching between multiple cameras are prevalent in real-world videos, and is not well addressed in previous works like IKEA-Manual, which estimated intrinsic separately for each part and will cause unrealistic relative poses in 3D space. We updated the website to include some examples of poses before and after refinement.We further develop an interactive interface to verify annotated poses in 3D, rather than relying solely on 2D renderings. This interface greatly improves the accuracy of the pose annotations.
>
> The diversity captured by our dataset (36 objects, ~ 90 environments) is thanks to the utilize of real-world videos. Our interface enables future works to achieve similar diversity efficiently, without extensive hardware/platform setups.
>
> The data collection interface is iterated over more than 4 months for its ease of use, accuracy, and efficiency. We will open-source our data collection and verification interface for future research.
>
>
> ### **Contribution w.r.t. IKEA-Manual**
>
> Our dataset differs from IKEA-Manual in three key aspects:
> 1. We provide detailed 3D motion information throughout furniture assembly.
> 2. We capture assembly in diverse real-world environments.
> 3. We propose a distinct part assembly task based on 2D video information in Section 5.4.
>
> Despite benchmarking the same part segmentation and part pose estimation tasks as IKEA-Manual, our experiment results reveal significant challenges of visual grounding on real-world images  (see attached PDF Figure A12-16) instead of on manual images that aim to clearly illustrate the assembly process.
>
> The part assembly task in our paper is different from the one in IKEA-Manual because our part assembly is according to the temporal information from videos. In contrast, IKEA-Manual only evaluates part assembly from 3D parts, ignoring information from the manual images.
>
> By providing dense spatio-temporal annotations on Internet videos, our dataset offers opportunities for studying different aspects of part assembly. Subtasks that can be evaluated on our dataset, such as 3D object tracking in videos and spatio-temporal action localization in videos. We invite the community to develop methods and benchmarking tasks using our dataset.
>
>
> ### **Scalability of the annotation pipeline.**
>
>
> While collecting such a complex dataset is challenging, we have strived for scalability in the design of our data collection pipeline. We developed a reusable interface that significantly reduces setup time compared to previous methods. Our approach allows for efficient collection of 6D pose trajectories for furniture assembly across diverse real-world settings, surpassing the environmental variety of datasets like HA-ViD and IKEA Ego 3D without the need for specialized hardware or controlled lab environments.
>
> Of course, human annotations remain necessary given the difficulty of recovering perfect 3D pose from depth, as shown in Figures A12-16 in the attached PDF. However, we believe that we have already achieved greater scalability than relevant datasets in terms of object and environment diversity. Our method captured 36 objects across more than 90 environments, far exceeding the variety in comparable datasets (Figure A17 and A11 in the attached PDF).
>
>
>
> ### **Additional baselines**
> Thank you for the suggestion.  While our main contribution is the dataset itself, we have: 1) included two additional baselines for pose estimation and 2) provided more insights in a detailed error analysis.
>
> We added two pose estimation methods based on differentiable rendering [A2]. The first method uses Mean Squared Error (MSE) loss, while the second additionally incorporates an occlusion-aware silhouette re-projection loss proposed in PHOSA [A1]. Both methods generate 20 random initial poses, refine the top five candidates with the lowest initial loss for 500 epochs, and selects the pose with the lowest final loss. We show results in the table below.
>
> |Method | ADD | ADD-S|
> |---|---|---|
> |Differential Rendering|3.33|2.91|
> |Differential Rendering (Occlusion Aware Loss)|3.29|2.86|
>
> Furthermore, we add a detailed error analysis in the attached PDF (Figure A12-16) that provides insights on current methods' performance. Our error analysis reveals that existing perception methods fail due to challenges associated with partial visibility, occlusions, ambiguous views, and diverse viewpoints. These failure modes reflect the complexities of real-world furniture assembly tasks captured by our dataset and suggest directions for future research. See the end of our website for more examples of real-world complexity.
>
> [A1] Zhang, J.Y., et al., 2020. Perceiving 3d human-object spatial arrangements from a single image in the wild. In: ECCV 2020. Springer, pp. 34-51.
>
> [A2] Ravi, N., et al., 2020. Accelerating 3d deep learning with pytorch3d. arXiv:2007.08501.
>
> ### **Typos**
> Thank you for pointing these out. We have fixed them in our paper.

---

> ### Author Rebuttal · Authors · 2024-08-17
>
> | Dataset                  | # Object Category | # Objects | Object Type                | Video Source  | # Environments | Environment Type | # Assemblers |
> |--------------------------|-------------------|-----------|----------------------------|---------------|-----------------|-------------------|--------------|
> | IKEA Video Manual (Ours) | 6                 | 36        | Furniture                  | Internet      | Multiple(~90)   | Indoor/Outdoor, Fixed/Moving Camera, First-/Third-Person View | Multiple |
> | Assembly101 [A3]              | 15 toy vehicle    | 101       | Toy vehicles               | Lab Collected | 1               | Lab, 8 Fixed + 4 Egocentric Cameras | 53 adults (28 males, 25 females) |
> | HA-ViD [A4]                   | 1                 | 35 parts  | Generic assembly box parts | Lab Collected | 1               | Lab, 3 Fixed Cameras | 30 participants (15 male, 15 female) |
> | IKEA-Manual [A5]              | 6                 | 102       | Furniture                  | -             | -               | - | - |
> | IKEA Ego 3D [A6]             | 4                 | 4         | Furniture                  | Lab Collected | 1               | Lab, Egocentric | 2 |
> | IKEA ASM [A7]                 | 3                 | 4         | Furniture                  | Lab Collected | 5               | Lab and home environments | 48 human subjects |
> | IKEA in the Wild (IAW) [A8]   | 14                | 420       | Furniture                  | Internet      | Multiple        | Indoor/Outdoor, Fixed/Moving Camera, First-/Third-Person View | Multiple |
>
>
> | Dataset                  | Image Modality   | # Manuals | 3D Object Model | 2D Info      | 3D Info    | Object Tracking | Camera Information |
> |--------------------------|-------------------|-----------|-----------------|--------------|------------|-----------------|---------------------|
> | IKEA Video Manual (Ours) | RGB               | 36        | ✓               | Segmentation | 6-DoF Pose | ✓               | Estimated camera, consistent throughout video segment |
> | Assembly101 [A3]               | RGB + Monochrome  | 0         | x               | x            | Depth      | x               | Calibrated multi-view cameras |
> | HA-ViD [A4]                   | RGB + Depth       | 0         | ✓               | Bounding Box | Depth      | ✓               | Calibrated multi-view cameras |
> | IKEA-Manual [A5]             | -                 | 102       | ✓               | Segmentation | 6-DoF Pose | -               | Estimated camera, different between parts |
> | IKEA Ego 3D [A6]              | RGB + Depth       | 0         | x               | x            | Depth      | x               | Calibrated egocentric camera |
> | IKEA ASM [A7]                 | RGB + Depth       | 0         | x               | Both         | Depth      | ✓               | Calibrated multi-view cameras |
> | IKEA in the Wild (IAW) [A8]   | RGB               | 461       | x               | x            | x          | x               | Diverse, uncalibrated |
>
>
> | Dataset                  | # Action Labels | Human/Hand Pose | Limiting Factors for Extension | Year |
> |--------------------------|-----------------|-----------------|--------------------------------|------|
> | IKEA Video Manual (Ours) | 137 manual steps, 1120 substeps | x | Annotations | 2024 |
> | Assembly101 [A3]             | 1380 (fine-grained), 202 (coarse) | 3D hand pose | Objects, Camera Rig Setup, Participants | 2022 |
> | HA-ViD [A4]                 | 75 (primitive tasks), 219 (atomic actions) | 2D + 3D | Generic Assembly Box Design, Participants | 2023 |
> | IKEA-Manual [A5]              | 393 manual steps | x | 3D Models, Manual Creation, Annotations | 2022 |
> | IKEA Ego 3D [A6]              | 56 atomic action | x | Furniture, Participants | 2024 |
> | IKEA ASM [A7]                 | 33 atomic actions | x | Furniture, Participants | 2023 |
> | IKEA in the Wild (IAW) [A8]  | 15649 action labels (manual step) | x | Video Availability, Annotations | 2023 |
>
>
> [A3] Sener, F., et al., 2022. Assembly101: A large-scale multi-view video dataset for understanding procedural activities. In: CVPR. pp. 21096-21106.
>
> [A4] Zheng, H., Lee, R. and Lu, Y., 2024. HA-ViD: a human assembly video dataset for comprehensive assembly knowledge understanding. NeurIPS, 36.
>
> [A5] Wang, R., et al., 2022. Ikea-manual: Seeing shape assembly step by step. NeurIPS, 35, pp.28428-28440.
>
> [A6] Ben-Shabat, Y., et al., 2024. IKEA Ego 3D Dataset: Understanding furniture assembly actions from ego-view 3D Point Clouds. In: WACV. pp. 4355-4364.
>
> [A7] Ben-Shabat, Y., et al., 2021. The ikea asm dataset: Understanding people assembling furniture through actions, objects and pose. In: WACV. pp. 847-859.
>
> [A8] Zhang, J., et al., 2023. Aligning step-by-step instructional diagrams to video demonstrations. In: CVPR. pp. 2483-2492.

---

> > ### Author Rebuttal · Authors · 2024-08-24
> >
> > Thank you again for your feedback. To address your comments, we have added a video segmentation experiment and provided an error analysis of this task to highlight the unique challenges. We have added more visualizations to our [website](https://yunongliu1.github.io/ikea-video-manual/) to highlight the diverse environments and our efforts to ensure annotation accuracy. Please see the general response under the title ***Additional Experiments and Visualizations*** for details. As the end of the discussion period is approaching, we want to make sure we have addressed all of your concerns. Please let us know if you have any additional comments.

---

> ### Author Response · Authors · 2024-08-27
> **Seeking Final Feedback**
>
> Dear Reviewer hZeh, thank you for your earlier feedback. We believe we have addressed your comments in our previous responses with additional experiments, visualizations, and analyses. As the discussion period is nearing its end, we wanted to check if you have any additional comments. We look forward to your feedback.

---

### Official Review · Reviewer_fTbB · 2024-07-22
**Well-motivated but the contributions can be enhanced further.**

**Rating:** 7
**Confidence:** 4
**Correctness:** Yes.
**Clarity:** Yes.

**Review:**

The authors extended an existing dataset by integrating 3D models of IKEA furniture parts, instructional manuals, and real-world assembly videos. I have no doubts about the motivation for this work. Indeed, to better understand assembly videos, we need to explore 4D grounding. However, I have some reservations regarding the contributions of this work.
1. The authors claim this is a large-scale dataset, but it only includes 98 videos, 34441 annotated frames. It's hard to consider this a large-scale dataset. The limited size makes it challenging to convincingly demonstrate its strong practical value.
2. The diversity of assemblies included in the dataset is insufficient. This undermines one of the key contributions claimed by the authors: "A novel multimodal dataset that captures the complexity of real-world furniture assembly tasks." The lack of diversity is evident in two aspects:
    ◦ The number of parts and assembly steps included is not high.
    ◦ In your dataset, are the assembly sequences the same for each furniture? If so, this does not reflect the variability seen in real-world scenarios. In real-world, especially in human assembly processes, there is significant variation in sequences, as shown in the existing assembly datasets, such as Assembly101 and HA-ViD.
3. The experimental design is relatively simple. First, in the assembly plan generation task, you seem to generate the assembly plan from 3D parts, whereas the significance of your work should be in generating the assembly plan directly from videos. Secondly, in the part-conditioned segmentation and part-conditioned pose estimation tasks, only testing pre-trained models does not fully demonstrate the value of constructing this dataset. Understanding the formation process of subassemblies is a key aspect of this dataset, but these tasks do not reflect how algorithms perform during the progression.
4. In the Shape Assembly with Instruction Videos experiment, what is the motivation behind the design of the second method? What is its significance?
5. From Figure 1, it seems your experimental design includes descovering the mapping from (b) to (c), to (d), and to (e), but why are there no experiments involving mapping from (a) to (b), such as action (assembly step) segmentation, localization, and recognition tasks?
6. The discussion on the limitations of this work is insufficient. I recommend discussing the limitations from more perspectives, such as data size, data modality, model development, application.

**Strengths:**

1. Novelty and Scope: The IKEA Video Manuals dataset addresses a critical gap in existing shape assembly datasets by providing detailed spatio-temporal alignments between assembly instructions, 3D models, and real-world videos.
2. Comprehensive Annotations: The dataset includes extensive annotations such as 2D-3D part correspondences, temporal step alignments, and part segmentation, which are valuable for various computer vision and robotics tasks.

**Additional Feedback:**

I have no more comment.

**Documentation:**

Yes.

**Ethics:**

No.

**Limitations:**

Yes.

**Opportunities For Improvement:**

1. Expand the Dataset Size: Increase the number of videos and annotated frames to enhance the dataset's scale and practical value.
2. Enhance Diversity: Increase the variety of parts, tasks, and assembly sequences included in the dataset to better reflect the complexity of real-world furniture assembly tasks.
3. Improve Experimental Design: Make your experiments closer to the real-world applications.
4. Discuss Limitations Thoroughly: Expand the discussion on the limitations of the work.

**Relation To Prior Work:**

Yes.

**Summary And Contributions:**

This paper introduces the IKEA Video Manuals dataset, a multimodal resource that combines 3D models of IKEA furniture parts, instructional manuals, and real-world assembly videos. The primary contribution is the dense spatio-temporal alignment of these diverse modalities. This dataset facilitates four essential tasks in shape assembly: assembly plan generation, part-conditioned segmentation, part-conditioned pose estimation, and video-based shape assembly. Its detailed annotations and alignment of instruction steps with video frames make it a valuable asset for advancing assembly video understanding.

---

> ### Author Rebuttal · Authors · 2024-08-17
>
> ### **Dataset scale**
>
> The value of our dataset lies in the dense spatio-temporal annotations on real-world assembly videos and the better diversity of objects and environments.
>
> As shown in our comparison table and figures (Figure A17 and A11 in the attached PDF), we uniquely provide 6D pose annotations on internet videos. In addition, we cover 36 distinct furniture items with 268 individual parts across 6 categories, significantly surpassing the variety found in existing furniture assembly video datasets. For context, HA-ViD includes one assembly apparatus with 35 parts, IKEA ASM includes 4 furniture items and Ego 3D includes 4 furniture items. Our dataset also presents 90 different real-world environments. This is in stark contrast to other datasets that typically feature only 1–5 controlled lab environments.
>
>
> ### **The diversity of assembly processes.**
>
> 1) Our dataset includes 36 furniture types with 268 parts across 6 categories, offering 137 manual steps and 1,120 substeps. This diversity exceeds other furniture assembly datasets, including IKEA ASM (4 furniture types across 3 categories) and Ego 3D (4 furniture types across 4 categories), as shown in our comparison table and figures. (Figure A17 and A11 in the attached PDF)
> 2) Our dataset captures variations in the assembly processes since they are extracted from Internet videos. 9 out of the 36 furniture objects have more than one unique assembly sequences. At most, the Laiva shelf in the dataset has 8 different assembly sequences.
>
>
>
> ### **Generating assembly plan from videos.**
>
> The formulation of our assembly plan generation task is based on videos, as discussed in the first sentence of Section 5.1. We evaluate two baselines without utilizing the videos because there are no established methods for extracting assembly plans for 3D parts from videos. A modular approach for solving this task could involve detecting key frames when two parts join and associating 3D parts with 2D image regions. These are exactly the two tasks that we individually evaluate in Section 5.4 and Section 5.2. By providing this diverse and realistic dataset and formulating these tasks, we hope to motivate the  community to study these problems.
>
>
> ### **Formation process of subassemblies.**
>
> We appreciate your recognition of our dataset's value in capturing video-based assembly processes. Our experiments address key components of grounding assembly instructions on videos, including part detection, pose estimation, and assembly order detection. Our dataset provides a foundation for developing sophisticated video-based models that can capture the progressive nature of assembly tasks. We believe our dataset will serve as a valuable resource for advancing research in this challenging area.
>
> ### **Design of the second method for shape assembly with instruction videos.**
>
> The motivation of using GPT-4o is to set a baseline performance for detecting key frames in furniture assembly videos, without specialized training. The inclusion of this baseline is significant because it represents the use of general-purpose vision-langauge models for video understanding tasks, specifically for temporal reasoning and localization (see Section 6.2 in [A1] for reference).
>
> [A1] Yang, Zhengyuan, Linjie Li, Kevin Lin, Jianfeng Wang, Chung-Ching Lin, Zicheng Liu, and Lijuan Wang. "The dawn of lmms: Preliminary explorations with gpt-4v (ision)." arXiv preprint arXiv:2309.17421 9, no. 1 (2023): 1.
>
> ### **Experiments involving mapping from instruction steps to videos**
> The mapping from (a) to (b) has been studied in the IAW (IKEA Assembly in the Wild) paper, which our work builds on. The IAW dataset already provides annotations and evaluations for action segmentation, localization, and recognition based on manual instructions. Our dataset's novel contribution lies in the dense spatio-temporal alignments between 3D models, instruction manuals, and videos (mappings b-c-d-e in Figure 1).
>
> ### **Discussion of limitations**
> Thank you for the suggestion. We expand our discussion below and will include it in the paper.
>
> Our work has several limitations. The dataset's limited size prevents large-scale training. The dataset currently only focuses on visual and 3D information; including other data modalities such as audio or textual data is an important future direction. The data collection process still requires manual annotation and verification, therefore presenting challenges for collecting data at a significantly larger scale. Current baselines demonstrate challenges in grounding 4D assembly but are yet to utilize this dataset for advanced model development. Since the dataset is focused on furniture assembly, whether models developed for this domain transfer to other assembly domains is left to be investigated. Future work could augment the dataset with additional modalities and develop algorithms leveraging instructional videos for 3D-grounded assembly plans. Broader implications include potential assistive technologies for individuals with disabilities, while internet-sourced data necessitates robust privacy and fair use methods.

---

> > ### Comment · Reviewer_fTbB · 2024-08-24
> >
> > Thank you for your response. I will keep my positive score.

---

> > > ### Author Rebuttal · Authors · 2024-08-24
> > >
> > > Thank you for recognizing the contribution of our dataset and maintaining a positive evaluation. We've added two baselines for video object segmentation. We also added more visualizations for the pose refinement process and the diverse environments on the [website](https://yunongliu1.github.io/ikea-video-manual/), further demonstrating our dataset's potential to understand the formation process of subassemblies. We look forward to any final feedback you may have.

---

> ### Author Rebuttal · Authors · 2024-08-17
>
> | Dataset                  | # Object Category | # Objects | Object Type                | Video Source  | # Environments | Environment Type | # Assemblers |
> |--------------------------|-------------------|-----------|----------------------------|---------------|-----------------|-------------------|--------------|
> | IKEA Video Manual (Ours) | 6                 | 36        | Furniture                  | Internet      | Multiple(~90)   | Indoor/Outdoor, Fixed/Moving Camera, First-/Third-Person View | Multiple |
> | Assembly101 [A2]              | 15 toy vehicle    | 101       | Toy vehicles               | Lab Collected | 1               | Lab, 8 Fixed + 4 Egocentric Cameras | 53 adults (28 males, 25 females) |
> | HA-ViD [A3]                   | 1                 | 35 parts  | Generic assembly box parts | Lab Collected | 1               | Lab, 3 Fixed Cameras | 30 participants (15 male, 15 female) |
> | IKEA-Manual [A4]              | 6                 | 102       | Furniture                  | -             | -               | - | - |
> | IKEA Ego 3D [A5]             | 4                 | 4         | Furniture                  | Lab Collected | 1               | Lab, Egocentric | 2 |
> | IKEA ASM [A6]                 | 3                 | 4         | Furniture                  | Lab Collected | 5               | Lab and home environments | 48 human subjects |
> | IKEA in the Wild (IAW) [A7]   | 14                | 420       | Furniture                  | Internet      | Multiple        | Indoor/Outdoor, Fixed/Moving Camera, First-/Third-Person View | Multiple |
>
>
> | Dataset                  | Image Modality   | # Manuals | 3D Object Model | 2D Info      | 3D Info    | Object Tracking | Camera Information |
> |--------------------------|-------------------|-----------|-----------------|--------------|------------|-----------------|---------------------|
> | IKEA Video Manual (Ours) | RGB               | 36        | ✓               | Segmentation | 6-DoF Pose | ✓               | Estimated camera, consistent throughout video segment |
> | Assembly101 [A2]               | RGB + Monochrome  | 0         | x               | x            | Depth      | x               | Calibrated multi-view cameras |
> | HA-ViD [A3]                   | RGB + Depth       | 0         | ✓               | Bounding Box | Depth      | ✓               | Calibrated multi-view cameras |
> | IKEA-Manual [A4]             | -                 | 102       | ✓               | Segmentation | 6-DoF Pose | -               | Estimated camera, different between parts |
> | IKEA Ego 3D [A5]              | RGB + Depth       | 0         | x               | x            | Depth      | x               | Calibrated egocentric camera |
> | IKEA ASM [A6]                 | RGB + Depth       | 0         | x               | Both         | Depth      | ✓               | Calibrated multi-view cameras |
> | IKEA in the Wild (IAW) [A7]   | RGB               | 461       | x               | x            | x          | x               | Diverse, uncalibrated |
>
>
> | Dataset                  | # Action Labels | Human/Hand Pose | Limiting Factors for Extension | Year |
> |--------------------------|-----------------|-----------------|--------------------------------|------|
> | IKEA Video Manual (Ours) | 137 manual steps, 1120 substeps | x | Annotations | 2024 |
> | Assembly101 [A2]             | 1380 (fine-grained), 202 (coarse) | 3D hand pose | Objects, Camera Rig Setup, Participants | 2022 |
> | HA-ViD [A3]                 | 75 (primitive tasks), 219 (atomic actions) | 2D + 3D | Generic Assembly Box Design, Participants | 2023 |
> | IKEA-Manual [A4]              | 393 manual steps | x | 3D Models, Manual Creation, Annotations | 2022 |
> | IKEA Ego 3D [A5]              | 56 atomic action | x | Furniture, Participants | 2024 |
> | IKEA ASM [A6]                 | 33 atomic actions | x | Furniture, Participants | 2023 |
> | IKEA in the Wild (IAW) [A7]  | 15649 action labels (manual step) | x | Video Availability, Annotations | 2023 |
>
>
> [A2] Sener, F., Chatterjee, D., Shelepov, D., He, K., Singhania, D., Wang, R. and Yao, A., 2022. Assembly101: A large-scale multi-view video dataset for understanding procedural activities. In Proceedings of the IEEE/CVF Conference on Computer Vision and Pattern Recognition (pp. 21096-21106).
>
> [A3] Zheng, H., Lee, R. and Lu, Y., 2024. HA-ViD: a human assembly video dataset for comprehensive assembly knowledge understanding. Advances in Neural Information Processing Systems, 36.
>
> [A4] Wang, R., Zhang, Y., Mao, J., Zhang, R., Cheng, C.Y. and Wu, J., 2022. Ikea-manual: Seeing shape assembly step by step. Advances in Neural Information Processing Systems, 35, pp.28428-28440.
>
> [A5] Ben-Shabat, Y., Paul, J., Segev, E., Shrout, O. and Gould, S., 2024. IKEA Ego 3D Dataset: Understanding furniture assembly actions from ego-view 3D Point Clouds. In Proceedings of the IEEE/CVF Winter Conference on Applications of Computer Vision (pp. 4355-4364).
>
> [A6] Ben-Shabat, Y., Yu, X., Saleh, F., Campbell, D., Rodriguez-Opazo, C., Li, H. and Gould, S., 2021. The ikea asm dataset: Understanding people assembling furniture through actions, objects and pose. In Proceedings of the IEEE/CVF Winter Conference on Applications of Computer Vision (pp. 847-859).
>
> [A7] Zhang, J., Cherian, A., Liu, Y., Ben-Shabat, Y., Rodriguez, C. and Gould, S., 2023. Aligning step-by-step instructional diagrams to video demonstrations. In Proceedings of the IEEE/CVF Conference on Computer Vision and Pattern Recognition (pp. 2483-2492).

---

### Official Review · Reviewer_U61q · 2024-07-24
**Review of IKEA Manuals at Work: 4D Grounding of Assembly Instructions on Internet Videos**

**Rating:** 7
**Confidence:** 4
**Clarity:** The paper is well written.

**Review:**

## Quality:
The paper presents a new dataset, the IKEA Video Manuals. This dataset is designed with multiple levels from top to bottom, and contains rich multimodal information. It implements step-by-step instructions on a video-based foundation, with spatial-temporal alignments between the different modalities and levels, and it captures the full complexity and diversity of real-world assembly processes. During annotation, the authors ensure the high quality and accuracy of the data by methods such as comparing the real-time 3D view with corresponding video frames, estimating camera parameters. The experiments conducted in assembly plan generation, part-conditioned segmentation, and part-conditioned pose estimation demonstrate that this dataset provides more comprehensive information, posing challenges to existing models. The experiment conducted in shape assembly with instruction videos proves the effectiveness of introducing video-based cues into the dataset.

## Clarity:
I think the paper is well-written with clear descriptions of the dataset's definition, structure, and annotation process. The introduction of background knowledge related to the work is highly relevant and concise. The methodology for utilizing the dataset in furniture assembly tasks is clearly explained, making it accessible for readers with related background.

## Originality:
The author still annotates based on the IKEA-Manual dataset and the IAW dataset to construct the dataset, and the tools used for constructing the dataset might not be too original. However, the proposed video-based idea is good, and it integrates detailed step-by-step guidance and 6-DoF pose annotations.

## Significance:
This work is significant to me, as it provides a valuable resource for advancing research in constructing complex 3D structures that require understanding of shape assembly tasks. The dataset's potential applications in developing assistive technologies for individuals with disabilities highlight its societal impact.

## Pros:
-  Provides a comprehensive dataset with 36 unique IKEA furniture models and 98 RGB video, contains 137 high-level assembly steps from instructional manuals and 1,120 detailed substeps from videos.
- Novel integration of video-based cues with 3D part models to facilitate real-world assembly tasks. Comprehensive annotations including substep instructions design and 2D-3D part correspondences.
- Detailed annotations including 34,441 annotated frames, detailed 6-DoF poses, and alignment with instructional manuals. It designed experiments to demonstrate the utility of the annotated data.
- About written, overall it has clear logic, and possesses an excellent writing style.

## Cons:
- The dataset relies on manual annotations, which may limit scalability and introduce subjectivity.
- The creation and maintenance of such a dataset require more resources, which might be a barrier for continuous updates or expansions.
- the dataset covers only six furniture categories, which may not fully represent the complexity found in more diverse or unconventional real-world types.
- A typo occurred in line 43 of the description of contributions: ‘; and,’.

**Strengths:**

1. From a video-based perspective, this dataset offers detailed step-by-step instructions and 6-DoF pose annotations, providing unprecedented depth and detail for learning assembly tasks from videos. It also addresses the limitations in capturing real-world complexity to a certain extent in such datasets. Moreover, the design of this dataset shows promising prospects for enhancing the autonomy of people with disabilities.

2. The dataset includes detailed annotations of 34,441 video frames, covering part identification and location, as well as the 3D pose of parts. The authors' methodological approach ensures high-quality data, such as accurately estimating video camera parameters. This high-quality and fine-grained annotation provides a foundation for developing more precise visual recognition and spatial reasoning algorithms. The experimental design mentioned in the paper reflects a thorough consideration of the dataset's practical utility, verifying its potential applications through the detection of key assembly steps and the construction of advanced assembly plans.

**Additional Feedback:**

1. Can you describe the details of locating individual 3D furniture parts?
2. Hope to study related models or methods to optimize annotations and curations, making it easier to expand to multiple types of items in various practical fields in the future.
3. Correct the writing error on line 43 of the contributions: ‘; and,’.

**Correctness:**

The claims made in the submission are correct. And the dataset is constructed in a sound way.

**Documentation:**

The submission provides sufficient details on data collection, organization, availability, and maintenance, ensuring ethical and responsible use. These details are documented in the paper and the appendix, including a URL for reviewer access and a clear hosting, licensing, and maintenance plan.

**Ethics:**

I didn’t suspect there were any ethical concerns with the submission that warrant further discussion or review.

**Limitations:**

I think the authors have adequately addressed the limitations and potential negative societal impacts of their work, demonstrating transparency and responsibility in their research approach.

**Opportunities For Improvement:**

- Still confined to IKEA-Manual dataset and the IAW dataset, and creating such datasets requires a lot of manual annotations and curations, which is not conducive to future expansions.
-   To avoid several types of ambiguities mentioned in the appendix, the author manually annotates the first frame for each substep after watching the entire video. How can the accuracy of manual annotations by the annotator be ensured? How should it be annotated if some parts do not appear in the first frame of a substep or are not clearly visible throughout the substep?

**Relation To Prior Work:**

It’s clearly discussed.

**Summary And Contributions:**

This work aims to provide a better dataset for the shape assembly task, facilitating researchers to develop assembly benchmarks. The paper introduces the IKEA Video Manuals dataset, a comprehensive resource that aligns 3D models of furniture parts with step-by-step assembly instructions, and real-world video demonstrations. This dataset possesses extensive spatial-temporal annotations, including 2D-3D part correspondences and part segmentations. Experiments on the dataset highlight significant challenges in grounding instructional assembly videos. Its contributions are as follows:
1. A novel multimodal dataset that captures the complexity of real-world furniture assembly tasks;
2. Comprehensive annotations, including 2D-3D part correspondences, temporal step alignments and so on.
3. Experiments on assembly plan generation, part segmentation, pose estimation, and part assembly based on videos to demonstrate the utility of the annotated data.

---

> ### Author Rebuttal · Authors · 2024-08-17
>
> ### **Scalability of the data collection procedure**
> While collecting such a complex dataset is challenging, we have strived for scalability in the design of our data collection pipeline. We developed a reusable interface that significantly reduces setup time compared to previous methods. Our approach allows for efficient collection of 6D pose trajectories for furniture assembly across diverse real-world settings, surpassing the environmental variety of datasets like HA-ViD and IKEA Ego 3D without the need for specialized hardware or controlled lab environments.
>
> Of course, human annotations remain necessary given the difficulty of recovering perfect 3D poses from depth, as shown in Figures A12-16 in the attached PDF. However, we believe that we have already achieved greater scalability than relevant datasets in terms of object and environment diversity. Our method captured 36 objects across more than 90 environments, far exceeding the variety in comparable datasets.
>
>
>
>
> ### **Complexity of the assembly processes**
> While our dataset covers six high-level furniture categories, it includes 36 distinct furniture types with 268 individual parts, significantly surpassing the diversity of existing datasets as shown in our comparison table below and figures (Figure A17 and A11 in the attached PDF). We capture these objects in over 90 different real-world environments, far exceeding the 1–5 controlled settings in other datasets. Importantly, our new Figure shows that 9 out of 36 furniture items have multiple assembly sequences, reflecting real-world variability. This combination of diverse furniture types, parts, environments, and assembly sequences provides a robust foundation for studying complex, real-world assembly processes, representing a significant advancement in the field.
>
>
> ### **Accuracy of the annotation**
>
> We appreciate the concern. Our annotation process ensures high accuracy and consistency, particularly in challenging scenarios.
>
> >How can the accuracy of manual annotations by the annotator be ensured?
>
> **How do we assign an identity to each part?**
>
> To ensure accurate part identification, we assign unique IDs to 3D model parts, matching the IKEA-Manual dataset. Annotators watch the entire video before starting, determining part IDs based on 3D model correspondence. For similar parts (e.g., table legs), we reference IKEA-Manual's assembly order. If leg_3 is first in IKEA-Manual, we label the first assembled leg as 3 in our video, with other similar parts identified by their relative positions.
>
> **How can accurate and consistent annotation be ensured?**
>
> To maintain annotation accuracy and consistency, our interface requires annotators to focus on one part throughout the video before moving to the next. We conduct multiple rounds of checks and re-annotations, with pose annotations beginning only after the mask is verified. Poses are initialized from previous frames to maintain consistency.
> This comprehensive approach ensures reliable part tracking, even for visually similar components, throughout the assembly process.
>
> >How should it be annotated if some parts do not appear in the first frame of a substep?
>
> This scenario doesn't occur in our dataset. We create new substeps whenever a new part appears or a sub-assembly is formed. This approach ensures all relevant parts or sub-assemblies are visible in the first frame of each substep.
>
> >How should it be annotated if some parts are not clearly visible throughout the substep?
>
> If a part and all its connected components (i.e., the whole sub-assembly) disappear after being visible in the first frame, it will still be kept in the assembly process. However, we mark its mask and pose as "not visible." There are no instances in our dataset where a part remains unclear throughout an entire substep.
>
> ### **Details on locating 3D parts**
> We first segment each video into substeps when a new part appears or a new subassembly is formed. For the first frame of each substep, we annotate part identities that are consistent throughout the video. The annotators then annotate masks for each part across all frames, which are verified for consistency manually. After mask verification, we proceed to 3D pose annotation with a custom interface that allowed annotators to view the parts from multiple angles to ensure accurate relative poses. We pay particular attention to coplanarity, inter-part distances, and correct relative locations (right/left, front/back, up/down). This multi-step approach, detailed in Sections 4.2-4.4 of our paper, allows us to accurately locate and pose 3D furniture parts in real-world assembly videos.
>
>
> ### **Extending to other domains.**
> We appreciate this suggestion for future work. Our current annotation pipeline is designed with flexibility in mind, and can be easily sped up when better video models are available. We are continuously working on that. Our open-sourced tools will enable the research community to contribute to these improvements.
>
>
> ### **Typo**
> Thank you for pointing these out. We have fixed them in our paper.

---

> ### Author Rebuttal · Authors · 2024-08-17
>
> | Dataset                  | # Object Category | # Objects | Object Type                | Video Source  | # Environments | Environment Type | # Assemblers |
> |--------------------------|-------------------|-----------|----------------------------|---------------|-----------------|-------------------|--------------|
> | IKEA Video Manual (Ours) | 6                 | 36        | Furniture                  | Internet      | Multiple(~90)   | Indoor/Outdoor, Fixed/Moving Camera, First-/Third-Person View | Multiple |
> | Assembly101 [A1]              | 15 toy vehicle    | 101       | Toy vehicles               | Lab Collected | 1               | Lab, 8 Fixed + 4 Egocentric Cameras | 53 adults (28 males, 25 females) |
> | HA-ViD [A2]                   | 1                 | 35 parts  | Generic assembly box parts | Lab Collected | 1               | Lab, 3 Fixed Cameras | 30 participants (15 male, 15 female) |
> | IKEA-Manual [A3]              | 6                 | 102       | Furniture                  | -             | -               | - | - |
> | IKEA Ego 3D [A4]             | 4                 | 4         | Furniture                  | Lab Collected | 1               | Lab, Egocentric | 2 |
> | IKEA ASM [A5]                 | 3                 | 4         | Furniture                  | Lab Collected | 5               | Lab and home environments | 48 human subjects |
> | IKEA in the Wild (IAW) [A6]   | 14                | 420       | Furniture                  | Internet      | Multiple        | Indoor/Outdoor, Fixed/Moving Camera, First-/Third-Person View | Multiple |
>
>
> | Dataset                  | Image Modality   | # Manuals | 3D Object Model | 2D Info      | 3D Info    | Object Tracking | Camera Information |
> |--------------------------|-------------------|-----------|-----------------|--------------|------------|-----------------|---------------------|
> | IKEA Video Manual (Ours) | RGB               | 36        | ✓               | Segmentation | 6-DoF Pose | ✓               | Estimated camera, consistent throughout video segment |
> | Assembly101 [A1]               | RGB + Monochrome  | 0         | x               | x            | Depth      | x               | Calibrated multi-view cameras |
> | HA-ViD [A2]                   | RGB + Depth       | 0         | ✓               | Bounding Box | Depth      | ✓               | Calibrated multi-view cameras |
> | IKEA-Manual [A3]             | -                 | 102       | ✓               | Segmentation | 6-DoF Pose | -               | Estimated camera, different between parts |
> | IKEA Ego 3D [A4]              | RGB + Depth       | 0         | x               | x            | Depth      | x               | Calibrated egocentric camera |
> | IKEA ASM [A5]                 | RGB + Depth       | 0         | x               | Both         | Depth      | ✓               | Calibrated multi-view cameras |
> | IKEA in the Wild (IAW) [A6]   | RGB               | 461       | x               | x            | x          | x               | Diverse, uncalibrated |
>
>
> | Dataset                  | # Action Labels | Human/Hand Pose | Limiting Factors for Extension | Year |
> |--------------------------|-----------------|-----------------|--------------------------------|------|
> | IKEA Video Manual (Ours) | 137 manual steps, 1120 substeps | x | Annotations | 2024 |
> | Assembly101 [A1]             | 1380 (fine-grained), 202 (coarse) | 3D hand pose | Objects, Camera Rig Setup, Participants | 2022 |
> | HA-ViD [A2]                 | 75 (primitive tasks), 219 (atomic actions) | 2D + 3D | Generic Assembly Box Design, Participants | 2023 |
> | IKEA-Manual [A3]              | 393 manual steps | x | 3D Models, Manual Creation, Annotations | 2022 |
> | IKEA Ego 3D [A4]              | 56 atomic action | x | Furniture, Participants | 2024 |
> | IKEA ASM [A5]                 | 33 atomic actions | x | Furniture, Participants | 2023 |
> | IKEA in the Wild (IAW) [A6]  | 15649 action labels (manual step) | x | Video Availability, Annotations | 2023 |
>
>
> [A1] Sener, F., Chatterjee, D., Shelepov, D., He, K., Singhania, D., Wang, R. and Yao, A., 2022. Assembly101: A large-scale multi-view video dataset for understanding procedural activities. In Proceedings of the IEEE/CVF Conference on Computer Vision and Pattern Recognition (pp. 21096-21106).
>
> [A2] Zheng, H., Lee, R. and Lu, Y., 2024. HA-ViD: a human assembly video dataset for comprehensive assembly knowledge understanding. Advances in Neural Information Processing Systems, 36.
>
> [A3] Wang, R., Zhang, Y., Mao, J., Zhang, R., Cheng, C.Y. and Wu, J., 2022. Ikea-manual: Seeing shape assembly step by step. Advances in Neural Information Processing Systems, 35, pp.28428-28440.
>
> [A4] Ben-Shabat, Y., Paul, J., Segev, E., Shrout, O. and Gould, S., 2024. IKEA Ego 3D Dataset: Understanding furniture assembly actions from ego-view 3D Point Clouds. In Proceedings of the IEEE/CVF Winter Conference on Applications of Computer Vision (pp. 4355-4364).
>
> [A5] Ben-Shabat, Y., Yu, X., Saleh, F., Campbell, D., Rodriguez-Opazo, C., Li, H. and Gould, S., 2021. The ikea asm dataset: Understanding people assembling furniture through actions, objects and pose. In Proceedings of the IEEE/CVF Winter Conference on Applications of Computer Vision (pp. 847-859).
>
> [A6] Zhang, J., Cherian, A., Liu, Y., Ben-Shabat, Y., Rodriguez, C. and Gould, S., 2023. Aligning step-by-step instructional diagrams to video demonstrations. In Proceedings of the IEEE/CVF Conference on Computer Vision and Pattern Recognition (pp. 2483-2492).

---

> > ### Author Rebuttal · Authors · 2024-08-24
> >
> > We appreciate your positive feedback. We have added visualizations of the pose refinement process on our website, illustrating our high accuracy in annotating relative poses. We also included additional video experiments with SAM2 and Cutie. We added visualizations of failure cases for the video segmentation task along with visualization of all diverse real-world environments on the [website](https://yunongliu1.github.io/ikea-video-manual/). Please see the general response under the title ***Additional Experiments and Visualizations*** for details. As the end of the discussion period is approaching, we want to make sure we have addressed all of your concerns. Please let us know if you have any additional comments.

---

> ### Author Response · Authors · 2024-08-27
> **Seeking Final Feedback**
>
> Dear Reviewer U61q, thank you for your earlier feedback. We believe we have addressed your comments in our previous responses with additional experiments and visualizations. As the discussion period is nearing its end, we wanted to check if you have any additional comments. We look forward to your feedback.

---

### Official Review · Reviewer_pCx9 · 2024-07-24
**IKEA Manuals 4D = Ikea Manuals + Ikea in the wild**

**Rating:** 6
**Confidence:** 3
**Clarity:** Yes!

**Review:**

Reasoning about what an object is is different from how it was created. This dataset provides a good push towards the second objective, which is a next step in visual understanding given that modern models are better equipped to estimate the existing static 3D of something constructed.

This work contributes an assembly dataset that is less in-the-lab, this is great!

It seems that these videos and the manuals are just the combination of two existing datasets, the Ikea IAW dataset and the Ikea manuals dataset. New annotations are included.

The hope of this paper seems to be that by combining the IAW dataset and Ikea manuals with additional annotation of the assembly process, that methods would do better at visual understanding tasks on these objects. However, it seems that baselines still perform poorly. The combination and additional annotation for this setting is a useful and valuable contribution.

pros:
- extensive 3D annotation
- additional benchmarks for understanding assembly
- a suite of 4 new tasks as a benchmark

cons:
- no real new data beyond annotations (it seems?)
- baseline methods perform poorly
- baseline methods seem limited per task, it seems there are obvious other methods that would be worth comparing to, even if they aren't 3D conditional

quality:
- the paper is well written
- the dataset is linked to in the appendix and instructions and the data.json is provided

clarity:
- the paper could be improved by being more forthright about being basically a re-annotation of two existing datasets, as it is this lede is buried

originality:
- originality is limited as this is a re-annotation of existing data

significance:
- despite this being a re-annotation it is very useful! In order to learn how objects are assembled it is useful to the community to contribute more high-fidelity data.

**Strengths:**

Adding 3D trajectories and video to the task of assembly on top of the existing IKEA manuals dataset is useful for this next frontier in visual processing, which is in understanding more than just recognition and seeing further in understanding how objects were built.

**Additional Feedback:**

Figure 2 (c) seems unbelievable, I think there was a typo. It says distribution of the number of 2D part masks and 6D part pose annotations "per video frame".

That would suggest that one video 1300 annotations in a single frame? Potentially on average per frame? That seems like a lot.

**Correctness:**

Yes, mostly, except for a few potential typos about metrics (see additional feedback).

**Documentation:**

N/A

**Ethics:**

N/A unless there are limitations on the use of IKEA data and/or concerns about the ethics of using videos of people assembling IKEA furniture from the internet.

**Limitations:**

Existing works like the IKEA manual dataset and Assembly101 are similar to this work. The authors justify that their work uses paired 3D with steps which is different from prior works, however, with good depth estimation systems/3D shape estimations, one might imagine that 3D models are recoverable separately from needing the annotated action. However, it is helpful in moving towards this goal to have these annotations now.

The 6 categories of furniture are limited and closer to a lab setting in terms of the diversity of objects. Towards understanding construction, ideally this is a good start for learning how things are put together.

It would have been nice to see how methods trained on this data could attempt to assemble novel objects.

Also, the methods introduced do not perform well on some of the tasks, even given the extension of annotations used for training. This, as the authors mention, limits the applicability to simply scale the methods as they would also need this additional annotation.

**Opportunities For Improvement:**

The dataset has few baseline methods applied to the experiments of part-conditioned segmentation and part-conditioned 6D pose estimation. Although they use SAM-6D and a CAD conditioned segmentation model, they should additionally be able to compare to other methods and it would better reveal the usefulness of their method.

For example, I could imagine rendering the parts into a number of images and then using a non-3D conditioned model as a segmentation method. This comparison could reveal how important the 3D information actually is.

**Relation To Prior Work:**

Yes, it is quite related to the prior IKEA manuals and Assembly101. It adds in 3D trajectories for assembly.

**Summary And Contributions:**

This dataset is a combination of two existing datasets with new annotations. It is a combination of the IKEA manuals and IKEA in the wild datasets. It adds annotations that track the 3D trajectories of parts through time during assembly.

These videos and models include 98 assembly videos from 36 IKEA manuals across 6 furniture categories. ~34K frames are annotated. Annotations include temporal step alignment and part segmentations.

Experiments aim to cover four tasks:

1) assembly part generation

This uses baselines from IKEA Manual, SingleStep and GeoCluster, aiming to generate an assembly plan from a video.

2) part-conditioned segmentation

This uses a frame and a given sub-assembly as input to predict a pixel-wise segmentation mask for that sub-assembly

3) part-conditioned pose estimation

This uses a frame and a given sub-assembly as input for estimating a pose

4) shape assembly with instruction video

This holistic task uses the whole video and the parts needed and aims to estimate all the 6D poses of the parts needed to construct the final object

---

> ### Author Rebuttal · Authors · 2024-08-17
>
> ### **Baseline performance.**
> Our primary contribution is a novel multimodal dataset designed to benchmark existing methods for furniture assembly. We don't claim the baselines as our contribution, nor are they trained on our dataset.
>
> The limited performance of the state-of-the-art methods on our dataset rather highlights its importance -- we show unique challenges in grounding furniture assembly instructions on Internet vidoes. To illustrate this, we've included a detailed error analysis in the attached PDF (Figure A12-16) and added examples at the end of our website showcasing challenges specific to real-world videos, such as camera view changes, heavy occlusions, and diverse environments.
>
>
> ### **More Baselines and Comparison Non-3D Methods.**
> In response to suggestions, we added two pose estimation methods based on differentiable rendering [A2]. The first method uses Mean Squared Error (MSE) loss, while the second additionally incorporates an occlusion-aware silhouette re-projection loss proposed in PHOSA [A1]. Both methods generate 20 random initial poses, refine the top five candidates with the lowest initial loss for 500 epochs, and select the pose with the lowest final loss. We show results in the table below.
>
> |Method | ADD | ADD-S|
> |---|---|---|
> |Differential Rendering|3.33|2.91|
> |Differential Rendering (Occlusion Aware Loss)|3.29|2.86|
>
> Regarding "using a non-3D conditioned model as a segmentation method", the CNOS method already employs this approach by rendering multiple object views and matching them against 2D image proposals from SAM or FastSAM. Similarly, our MegaPose baseline renders multiple 3D object views for comparison with the input image using a learned 2D CNN.
>
> [A1] Ravi, N., Reizenstein, J., Novotny, D., Gordon, T., Lo, W.Y., Johnson, J. and Gkioxari, G., 2020. Accelerating 3d deep learning with pytorch3d. arXiv preprint arXiv:2007.08501.
>
> [A2] Zhang, J.Y., Pepose, S., Joo, H., Ramanan, D., Malik, J. and Kanazawa, A., 2020. Perceiving 3d human-object spatial arrangements from a single image in the wild. In Computer Vision–ECCV 2020: 16th European Conference, Glasgow, UK, August 23–28, 2020, Proceedings, Part XII 16 (pp. 34-51). Springer International Publishing.
>
>
> ### **Relations with related datasets.**
> Thank you for the suggestion. Our dataset uniquely provides 6D pose annotations on internet videos, capturing furniture assembly in diverse, real-world settings. We provide a comparison table to highlight the differences with existing works and will add this table to the paper. Assemby101 does not provide 3D information. IKEA-Manual dataset provides pose annotation for each part, without ensuring correct relative pose between parts and consistency between steps. We ensure this consistency through our new data annotation process, as discussed in Section 4.4. We also updated some examples of 6D poses before and after refinement on our website, showing the specific challenge when grounding 3D on images.
>
>
>
> ### **Recovering poses from depth estimation and 3D shape estimation.**
> Existing methods for depth estimation and 3D shape estimation are not robust enough to be applied to real-world videos; therefore, we opt for creating a data collection pipeline that involves human annotators and verifiers.
>
> Our added error analysis in the attached PDF (Figure A12-16) confirms that real-world assembly videos often feature occlusions, challenging viewpoints, and partial visibility. These are difficult cases for recovering poses directly from depth estimation.
>
>
>
> ### **Categories of furniture.**
> Our dataset includes 36 furniture types with 268 parts across 6 categories (see FigureA1 in the supplementary material). As shown in the comparison table and the comparison figures (Figure A17 and A11 in the PDF), our dataset greatly improves the diversity of objects and environments than existing assembly datasets with 3D information. Specifically, IKEA ASM includes 4 furniture items and Ego 3D includes 4 furniture items, HA-ViD includes one object with 35 parts. Our dataset also presents 90 different real-world environments. We updated the website to include an overview of the environments.This is in stark contrast to other datasets that typically feature only 1–5 controlled lab environments.
>
>
> ### **Data vs Annotations.**
> Thank you for the suggestion. The main contribution of our paper is a new multimodal dataset containing dense spatio-temporal annotations for real-world furniture assembly. We will revise the paper to be more explicit about this point.
>
>
> ### **Typo**
> Thank you for pointing this out. We will update the caption to fix this typo. Figure 2.c shows the distribution of the number of 2D part masks and 6D part pose annotations per video.

---

> > ### Author Response · Authors · 2024-08-27
> > **Seeking Final Feedback**
> >
> > Dear Reviewer pCx9, thank you for your earlier feedback. We believe we have addressed your comments in our previous responses with additional experiments and visualizations. As the discussion period is nearing its end, we wanted to check if you have any additional comments. We look forward to your feedback.

---

> ### Author Rebuttal · Authors · 2024-08-17
>
> | Dataset                  | # Object Category | # Objects | Object Type                | Video Source  | # Environments | Environment Type | # Assemblers |
> |--------------------------|-------------------|-----------|----------------------------|---------------|-----------------|-------------------|--------------|
> | IKEA Video Manual (Ours) | 6                 | 36        | Furniture                  | Internet      | Multiple(~90)   | Indoor/Outdoor, Fixed/Moving Camera, First-/Third-Person View | Multiple |
> | Assembly101 [A3]              | 15 toy vehicle    | 101       | Toy vehicles               | Lab Collected | 1               | Lab, 8 Fixed + 4 Egocentric Cameras | 53 adults (28 males, 25 females) |
> | HA-ViD [A4]                   | 1                 | 35 parts  | Generic assembly box parts | Lab Collected | 1               | Lab, 3 Fixed Cameras | 30 participants (15 male, 15 female) |
> | IKEA-Manual [A5]              | 6                 | 102       | Furniture                  | -             | -               | - | - |
> | IKEA Ego 3D [A6]             | 4                 | 4         | Furniture                  | Lab Collected | 1               | Lab, Egocentric | 2 |
> | IKEA ASM [A7]                 | 3                 | 4         | Furniture                  | Lab Collected | 5               | Lab and home environments | 48 human subjects |
> | IKEA in the Wild (IAW) [A8]   | 14                | 420       | Furniture                  | Internet      | Multiple        | Indoor/Outdoor, Fixed/Moving Camera, First-/Third-Person View | Multiple |
>
>
> | Dataset                  | Image Modality   | # Manuals | 3D Object Model | 2D Info      | 3D Info    | Object Tracking | Camera Information |
> |--------------------------|-------------------|-----------|-----------------|--------------|------------|-----------------|---------------------|
> | IKEA Video Manual (Ours) | RGB               | 36        | ✓               | Segmentation | 6-DoF Pose | ✓               | Estimated camera, consistent throughout video segment |
> | Assembly101 [A3]               | RGB + Monochrome  | 0         | x               | x            | Depth      | x               | Calibrated multi-view cameras |
> | HA-ViD [A4]                   | RGB + Depth       | 0         | ✓               | Bounding Box | Depth      | ✓               | Calibrated multi-view cameras |
> | IKEA-Manual [A5]             | -                 | 102       | ✓               | Segmentation | 6-DoF Pose | -               | Estimated camera, different between parts |
> | IKEA Ego 3D [A6]              | RGB + Depth       | 0         | x               | x            | Depth      | x               | Calibrated egocentric camera |
> | IKEA ASM [A7]                 | RGB + Depth       | 0         | x               | Both         | Depth      | ✓               | Calibrated multi-view cameras |
> | IKEA in the Wild (IAW) [A8]   | RGB               | 461       | x               | x            | x          | x               | Diverse, uncalibrated |
>
>
> | Dataset                  | # Action Labels | Human/Hand Pose | Limiting Factors for Extension | Year |
> |--------------------------|-----------------|-----------------|--------------------------------|------|
> | IKEA Video Manual (Ours) | 137 manual steps, 1120 substeps | x | Annotations | 2024 |
> | Assembly101 [A3]             | 1380 (fine-grained), 202 (coarse) | 3D hand pose | Objects, Camera Rig Setup, Participants | 2022 |
> | HA-ViD [A4]                 | 75 (primitive tasks), 219 (atomic actions) | 2D + 3D | Generic Assembly Box Design, Participants | 2023 |
> | IKEA-Manual [A5]              | 393 manual steps | x | 3D Models, Manual Creation, Annotations | 2022 |
> | IKEA Ego 3D [A6]              | 56 atomic action | x | Furniture, Participants | 2024 |
> | IKEA ASM [A7]                 | 33 atomic actions | x | Furniture, Participants | 2023 |
> | IKEA in the Wild (IAW) [A8]  | 15649 action labels (manual step) | x | Video Availability, Annotations | 2023 |
>
>
> [A3] Sener, F., Chatterjee, D., Shelepov, D., He, K., Singhania, D., Wang, R. and Yao, A., 2022. Assembly101: A large-scale multi-view video dataset for understanding procedural activities. In Proceedings of the IEEE/CVF Conference on Computer Vision and Pattern Recognition (pp. 21096-21106).
>
> [A4] Zheng, H., Lee, R. and Lu, Y., 2024. HA-ViD: a human assembly video dataset for comprehensive assembly knowledge understanding. Advances in Neural Information Processing Systems, 36.
>
> [A5] Wang, R., Zhang, Y., Mao, J., Zhang, R., Cheng, C.Y. and Wu, J., 2022. Ikea-manual: Seeing shape assembly step by step. Advances in Neural Information Processing Systems, 35, pp.28428-28440.
>
> [A6] Ben-Shabat, Y., Paul, J., Segev, E., Shrout, O. and Gould, S., 2024. IKEA Ego 3D Dataset: Understanding furniture assembly actions from ego-view 3D Point Clouds. In Proceedings of the IEEE/CVF Winter Conference on Applications of Computer Vision (pp. 4355-4364).
>
> [A7] Ben-Shabat, Y., Yu, X., Saleh, F., Campbell, D., Rodriguez-Opazo, C., Li, H. and Gould, S., 2021. The ikea asm dataset: Understanding people assembling furniture through actions, objects and pose. In Proceedings of the IEEE/CVF Winter Conference on Applications of Computer Vision (pp. 847-859).
>
> [A8] Zhang, J., Cherian, A., Liu, Y., Ben-Shabat, Y., Rodriguez, C. and Gould, S., 2023. Aligning step-by-step instructional diagrams to video demonstrations. In Proceedings of the IEEE/CVF Conference on Computer Vision and Pattern Recognition (pp. 2483-2492).

---

> > ### Author Rebuttal · Authors · 2024-08-24
> >
> > Thank you again for your feedback. We have updated our [website](https://yunongliu1.github.io/ikea-video-manual/) with new visualizations of the real-world environments for the six object categories, showing greater diversity than existing datasets. We have also added a new video segmentation experiment using two SOTA methods, SAM2 and Cutie. Please see the general response under the title ***Additional Experiments and Visualizations*** for details. As the end of the discussion period is approaching, we want to make sure we have addressed all of your concerns. Please let us know if you have any additional comments.

---

### Author Rebuttal · Authors · 2024-08-17

We would like to once again thank all the reviewers for their insightful comments and active engagement. Your feedback has been invaluable in improving our work.

We are encouraged that all reviewers have a positive view of our work. Specifically, they found。
* Our annotations are **extensive** and **high-quality**, including **detailed 3D trajectories** and **6-DoF pose** annotations (pCx9, U61q, fTbB).
* Our **experimental design** reflects a thorough consideration of the dataset's **practical utility** (U61q).
* Our dataset captures the **complexity of real-world** assembly processes **better than lab-based datasets** (pCx9, U61q).
* Our paper is **well-written**, clear, and presents a thorough consideration of the dataset's **practical utility** (pCx9, U61q, hZeh).
* Our work is **distinct from previous datasets** by requiring grounding in real-world images rather than manual images (hZeh).

We very much appreciate the reviewes. Based on their feedback, we have made the following improvements:
* Clarifying the novelty and contributions of our dataset, particularly the unique 4D groundings of assembly instructions in internet videos.
* Providing a comprehensive comparison with existing datasets to highlight our dataset's diversity and real-world complexity.
* Elaborating on our data collection and annotation process, addressing concerns about scalability and accuracy.
* Adding two more baselines for pose estimation based on differentiable rendering.
* Providing a detailed error analysis, drawing more insights on the challenges of grounding assembly in real-world videos.
* Conducting additional video segmentation experiments using SOTA methods (SAM2 and Cutie).
* Adding new visualizations of diverse environments and the pose refinement process on our project website.

As elaborated in each individual response, given more space (in the final version), we will be able to move the above clarifications and details to the main paper and thus further improve our draft.

---

> ### Author Rebuttal · Authors · 2024-08-24
>
> ### Additional Experiments and Visualizations
> We would like to thank all the reviewers for their insightful comments, which helped us greatly improve the paper. In the past few days, we have conducted additional experiments to evaluate the tracking of furniture parts in the assembly processes. We also created additional visualizations and analyses. We provide details below.
>
> 1. We evaluated two state-of-the-art video segmentation models, SAM2 [A1] and Cutie [A2], on our dataset. We closely follow the experiment design of standard video object segmentation, ensuring a fair comparison with other benchmark datasets. For each trial, we initialize a target part's mask with groundtruth in the first frame of the substep when the target part appears. We evaluate the performance on subsequent frames. We filter out video segments with less than 20 frames to ensure meaningful temporal evaluation. We list the performance of both baselines on our dataset and on related video segmentation datasets below.
>
> | Dataset | SAM2 (Hiera-L) J&F | Cutie-base J&F |
> |---------|---------------------|-----------------|
> | Our Dataset | 73.6 | 54.7 |
> | MOSE | 77.2 | 69.9 |
> | DAVIS 2017 val | 91.6 | 87.9 |
> | LVOS val | 76.1 | 66.0 |
> | SA-V val | 75.6 | 60.7 |
> | SA-V test | 77.6 | 62.7 |
> | YTVOS 2019 val | 89.1 | 87.0 |
>
> The results show that SAM2 performs worse on our dataset compared to existing benchmarks. Cutie experiences a more significant decrease in performance. SAM2's performance is around 4.7% lower than on MOSE. Cutie's performance drops by around 21.7%. Our error analysis reveals challenges such as abrupt camera movements, similar part appearances, occlusions, small parts, and extended assembly processes. These challenging cases, more prevalent in our dataset than in MOSE or SA-V, underscore the potential of our dataset in advancing video object segmentation research.
>
> 2. We've included additional examples to compare the annotated relative poses between furniture parts before and after our verification and reannotation process. These examples highlight the limitations of the traditional PnP annotation method and emphasize the crucial role of relative pose in 3D annotation, which our annotation interface uniquely focuses on.
>
> 3. To better illustrate the diversity of our dataset, we have added a visualization of all environments on our website under the section Environments (98 Videos).
>
> We hope the additional experiments and analyses confirm the novelty and significance of our contribution. We're prepared to provide additional details if required. Thank you!
>
> [A1] Ravi, N., Gabeur, V., Hu, Y.T., Hu, R., Ryali, C., Ma, T., Khedr, H., Rädle, R., Rolland, C., Gustafson, L. and Mintun, E., 2024. Sam 2: Segment anything in images and videos. arXiv preprint arXiv:2408.00714.
>
> [A2] Cheng, H.K., Oh, S.W., Price, B., Lee, J.Y. and Schwing, A., 2024. Putting the object back into video object segmentation. In Proceedings of the IEEE/CVF Conference on Computer Vision and Pattern Recognition (pp. 3151-3161).

---

### Decision · Program_Chairs · 2024-09-26

**Decision:**

Accept (Poster)

**Comment:**

The paper has been reviewed by four reviewers. After discussion, the reviewers’ recommendations are unanimously positive, with the following scores: “6: Marginally above acceptance threshold”, “7: Good paper, accept”, “7: Good paper, accept”, “6: Marginally above acceptance threshold”.

While not all reviewers participated in the discussion highlighted limitations such as limited baselines, scalability, scale and diversity, have been addressed by the authors in the rebuttal, and the AC is confident these improvements can be reflected in the final version of the manuscript.